# Straw application promotes soil carbon storage by affecting aggregate-associated bacterial community structure and RuBisCO activity: a 35-year field experiment

Xinyue Li,[1] Rong Huang,[1] Yong Wang,[2] Hong Jiang,[3] Youlin Luo,[1] Changquan Wang,[1] Bing Li[1]

**ABSTRACT** Soil bacterial community structure and carbon-fixation functional genes are easily influenced by straw application, especially in the microenvironment of the aggregate. Here, soil aggregate samples were collected from a 35-year rice–wheat rotation experiment field with the following treatments: no straw or chemical fertilizer (Ctrl); chemical fertilizer (NPK); and straw application + chemical fertilizer (NPKS). Compared with the control and NPK treatments, NPKS treatment enhanced the contents of soil organic carbon (SOC) and microbial biomass carbon (MBC) significantly in 0.25–1 mm aggregate soil. Similarly, higher ribulose-1,5-bisphosphate carboxylase/oxygenase (RuBisCO) activity (172.92 nmol $CO_2$ $g^{-1}$ soil $h^{-1}$) and abundance of *cbbL* (6.26 × $10^8$ copies $g^{-1}$) in the 0.25–1 mm aggregate fraction were observed under the NPKS treatment. Bacterial community diversity in various soil aggregate fractions decreased under the NPK treatment but increased under the NPKS treatment, compared with the control. The relative abundances of Proteobacteria and Actinobacteria in the 0.25–1 mm aggregate under the NPKS treatment were greater than those in the other treatments, indicating sufficient nutrient supply in this aggregate fraction, beneficial to eutrophic bacterial growth. Redundancy analysis showed that bacterial community composition strongly affected SOC and MBC distribution across soil aggregate fractions. Bacterial community contributed to RuBisCO activity by affecting the *cbbL* gene (path coefficient = 0.67, $P < 0.001$), which positively affected SOC content. In conclusion, the dominant soil aggregate fraction (0.25–1 mm) is mainly involved in carbon fixation in paddy, and straw application promoted carbon storage by affecting bacterial community structure in soil aggregates.

**IMPORTANCE** Paddy soils have been under frequent disturbance through field management activities (fertilization and straw application) for a long time, and carbon pool change is active and frequent. Soil aggregate plays an important role in carbon capture and storage. Variations in aggregate could alter microbial habitats and life strategies, thus triggering the renewal of soil organic carbon (SOC) encapsulated in aggregates. Straw application has both economic and ecological benefits, and it has been widely promoted in paddy fields with obvious effects. Here, a 35-year long-term positioning experiment was carried out to explore the mechanisms through which microbial communities increase SOC content as influenced by soil aggregate size. The findings enhance our understanding of carbon storage and aggregate-associated microbial mechanisms in paddy soil, in addition to facilitating the enhancement of paddy productivity and promoting the rational utilization of straw resources.

**KEYWORDS** straw return, soil aggregate, carbon fixation potential, *cbbL*, RuBisCO, bacterial community

**Peer Reviewer** Yudan Bai, East China Normal University, Shanghai, China

Address correspondence to Bing Li, benglee@163.com.

The authors declare no conflict of interest.

See the funding table on p. 14.

$S$oil organic carbon (SOC) plays a vital role in soil biological function and fertility maintenance (1). Nearly 90% of the SOC in the soil surface layer is encapsulated in different soil aggregate size fractions, which facilitates SOC stability by physical protection (2). Generally, soil aggregates can be divided into macroaggregates (>0.25 mm size) and microaggregates (<0.25 mm size) (3). Some reports have suggested that SOC in microaggregates, which has much greater physical protection, is less variable than that in macroaggregates (4). However, because macroaggregates have more abundant pore spaces and substrate availability, they may play a core role in SOC turnover processes compared with the smaller microaggregates (5). In addition, aggregate-associated SOCs are sensitive to agronomic management measures, such as straw application (6), which is a key source of organic material in soil and establishes favorable conditions for SOC decomposition and accumulation (7).

Paddy soils have been under frequent disturbance through field management activities (tillage, fertilization, irrigation, and so on) for a long time, and carbon pool change is active and frequent (8, 9). Therefore, investigating how agronomic management measures influence the transformation of aggregate-associated SOCs could facilitate the management of paddy fields and the rational use of straw to mitigate global climate change and increase soil quality.

Soil microorganisms have an irreplaceable role in the understanding of SOC sequestration and decomposition, and their community composition and diversity have profound impacts on rice–wheat rotation soil fertility and production. Previous reports suggest that agronomic measures affect SOC transformation, microbial communities, and enzyme activities, which interact and interconnect (10). Similarly, aggregate-associated SOCs are affected by microbial and enzyme activity (11). Variations in soil aggregates could alter microbial habitats and life strategies, thus triggering the renewal of SOCs encapsulated in aggregates (12).

Substrates in soil aggregates are reportedly a primary factor influencing microbial activity that drives SOC formation and decomposition (13, 14). High nutrient availability in macroaggregates is beneficial to microbial growth and hence accelerates SOC regeneration (15). Some studies have suggested that obscured carbon is protected in aggregates, minimizing decomposition by microorganisms (16). Therefore, carbon distribution among various aggregates is a result of microbial growth and activity. Differences in the aggregate environment based on particle and pore sizes may lead to changes in the internal SOC fixation mechanism, which needs to be explored further.

Autotrophic microorganisms could fix carbon dioxide ($CO_2$), mainly through the Calvin cycle pathway; thereafter, they synthesize sugars for absorption and utilization after a series of biochemical reactions (17). In particular, ribulose-1,5-bisphosphate carboxylase/oxygenase (RuBisCO) is the key enzyme of the Calvin cycle, indicative of $CO_2$ fixation, marked by its functional large-subunit gene (*cbbL*). RuBisCO occurs in three related forms (I, II, and III) that vary in structure, and analyses of form I RuBisCO offer the greatest insights for understanding soil autotrophic bacteria activity (17). To date, $CO_2$ emissions from paddy have become a research hotspot, and more attention should be paid to the carbon fixation by autotrophic microorganisms. Responses of *cbbL* and RuBisCO activity to straw application across aggregate fractions remain unclear. Microbial growth and function vary in different soil aggregate fractions due to various microenvironment conditions ascribed to oxygen contents, substrate availability, pH, pore characteristics, and water availability (18). In rice–wheat rotation systems, how microbial communities affect the SOC cycle in the aggregate microenvironment remains unclear, in addition to microbial responses to long-term straw return.

Considering the results of previous studies on changes in SOC and microbial distribution in aggregates, it is necessary to explore the effect of dynamic SOC balance on microbial function using long-term positioning experiments. We hypothesized that (i) straw return could alter microbial community structure by reorganizing the distribution of aggregate fractions, which constitute the main microbial habitat and (ii) the >0.25 mm aggregate fractions are the main sites of carbon fixation, with greater microbial biomass,

*cbbL* abundance, and RuBisCO activity, promoting carbon renewal. The findings enhance our understanding of carbon storage mechanisms in paddy soil, in addition to facilitating the enhancement of paddy productivity and promoting the rational utilization of straw resources.

## MATERIALS AND METHODS

### Experimental design

The field experimental site of the rice–wheat rotation was in Zhongjiang County (30°59′43″ N, 104°57′12″ E), Sichuan Province, China. The experimental site has been in use for paddy cultivation since 1984. This area has a shallow, hilly topography in the middle of the Sichuan Basin, with a subtropical monsoon humid climate. The average annual precipitation is 1,200 mm, and the average annual temperature is 16.7℃. The soil in the area developed from an alluvial-alluvial deposit of weathering between thinly bedded sands and thickly bedded shales of Lower Cretaceous soil, and it is classified as purple soil. The topsoil (0–20 cm) properties before the experiment included pH 7.60; 24.40 g kg$^{-1}$ soil organic matter, 1.46 g kg$^{-1}$ total nitrogen (N) content, 138.00 mg kg$^{-1}$ alkali-hydrolyzable N, 10.80 mg kg$^{-1}$ available P, 122.00 mg kg$^{-1}$ available potassium, and 15.38 cmol kg$^{-1}$ cation exchange capacity. The sand, silt, and clay contents were 35%, 44%, and 21%, respectively.

Three field experimental plots were established, including (i) Ctrl (control, no chemical fertilizer or straw) (ii), NPK (chemical fertilizer), and (iii) NPKS (straw return plus chemical fertilizer). The cropping system was an annual rice–wheat rotation. Similar amounts of compound fertilizer were applied to the NPK and NPKS plots. The amounts of chemical fertilizer applied in the rice season were 183.51 kg·hm$^{-2}$ N, 34.54 kg·hm$^{-2}$ P$_2$O$_5$, and 48.88 kg·hm$^{-2}$ K$_2$O, and the amounts applied in the wheat season were 132.23 kg·hm$^{-2}$ N, 34.54 kg·hm$^{-2}$ P$_2$O$_5$, and 48.88 kg·hm$^{-2}$ K$_2$O. All the chemical fertilizers were used as basal fertilizer and spread evenly on the soil surface 1 d before wheat sowing or rice transplanting. In the Ctrl and NPK plots, the aboveground straw was removed after crop harvest. In the NPKS plots, rice or wheat straw was crushed and incorporated directly by plowing into the field after each crop harvest. The amount of straw returned to the field was approximately 5,000 kg hm$^{-2}$ in the rice season and 3,000 kg hm$^{-2}$ in the wheat season.

### Soil sample collection and separation of soil aggregates

Soil was sampled after rice harvest in October 2019. Each experimental plot was divided into triplicate sampling areas of equal proportion. Soil surface impurities were scraped away, and five undisturbed soil samples (0–20 cm depth) were collected randomly from each sampling area and mixed homogeneously. The samples were then put in sterile rigid sampling boxes and placed immediately in an ice cooler. After removing visible stones and plant fragments, the soil was broken apart gently along the natural fracture planes to a size of approximately 5 mm.

Soil aggregates were separated using the modified dry-sieving method according to Wang et al. (19) and Bach et al. (20). In sterile conditions, the "optimal moisture" method was used for isolation of aggregates from fresh soil with a moisture content of 10%–15% to minimize disturbance to microbial communities. Three sieve nests (2, 1, and 0.25 mm pore diameters) were installed in turn, and bulk soils were vibrated mechanically to separate them into large macroaggregates (>2 mm), medium macroaggregates (1–2 mm), small macroaggregates (0.25–1 mm), and microaggregates (<0.25 mm). One part of the aggregate sample was conserved at 4℃ for RuBisCO activity and microbial biomass carbon (MBC) analyses. The second part was preserved at −80℃ for subsequent DNA extraction. The remaining soil was air-dried and passed through a 0.15 mm sieve for SOC analysis.

## Soil property analyses

SOC was measured by adding potassium dichromate and then performing external heating (10). MBC was treated with chloroform fumigation and then measured using a Total Organic Carbon Analyzer (Elementar GmbH, Hanau, Germany) (10). Soil RuBisCO activity was measured using the NADH-coupled spectrophotometric method (21, 22). First, the soil solution (50 µL) was extracted and then incubated at 30°C to restore enzyme activity. Ninety-six-hole microporous plates were used, and blank holes, standard holes, and sample holes were set up. The absorbance of the liquid was analyzed at 340 nm using an enzyme labeling instrument (Varioskan LUX, Thermo), and RuBisCO activity was expressed in nmol $CO_2$ $g^{-1}$ soil $h^{-1}$.

## DNA extraction and microbial community structure

Total DNA from 36 samples (three treatments, four aggregate sizes, repeated three times) was extracted using an E.Z.N.A. soil DNA kit (Omega Bio-Tek, Norcross, GA, USA). Final DNA concentration and purity were determined with a NanoDrop2000 spectrophotometer (Thermo Fisher Scientific, Wilmington, USA) and visualized with 1% agarose gel electrophoresis. Using a thermocycler PCR system (GeneAmp9700, ABI, Foster City, USA), bacterial 16S rRNA in the V3–V4 region was amplified using the primers 338F (5′-ACTCCTACGGGAGGCAGCAG-3′) and 806R (5′-GGACTACHVGGGTWTCTAAT-3′) (23). The amplification process was initial denaturation at 95°C for 3 min, followed by 27 cycles of denaturing at 95°C for 30 s, annealing at 55°C for 30 s, and extension at 72°C for 45 s, with a final extension at 72°C for 10 min. PCR products were extracted into 2% agarose gels and purified with an AxyPrep DNA Gel Extraction Kit (Axygen Biosciences, Union City, NJ, USA). PCR products were quantified using a QuantiFluor-ST (Promega, Fitchburg, MA, USA).

The Illumina MiSeq PE300 platform (Illumina Inc., San Diego, CA, USA) was used for sequencing according to the standard protocol of Majorbio Biopharmaceutical Technology Co., Ltd. (Shanghai, China). The original readings were deposited in the National Center for Biotechnology Information (NCBI) database, Sequence Read Archive (accession number PRJNA987044). Raw fastq files were quality-filtered and merged using Trimmomatic (https://github.com/usadellab/Trimmomatic) and FLASH v1.2.11 (https://bioweb.pasteur.fr/packages/pack@FLASH@1.2.11), respectively. Operational taxonomic units were clustered based on 97% similarity using UPARSE (v 7.0.109) (24). Each 16S rRNA gene sequence was classified using the RDP Classifier (v 2.11) algorithm (25), according to the Silva (v138) 16S rRNA database (https://www.arb-silva.de/).

## *cbbL* gene quantification

The large subunit of form I RubisCO is encoded by the cbbL gene (17). *cbbL* (I) was amplified on a fluorescence quantitative PCR instrument (ABI7300, Applied Biosystems, Foster City, CA, USA) with primers K2f (5′-ACCA[C/T]CAAGCC[G/C]AAGCT[C/G]GG-3′) and V2r (5′-GCCTTC[C/G]AGCTTGCC[C/G]ACC[G/A]C-3′) (26). According to the thermal protocol described by Zhou et al. (21), 20 µL of total reaction solution containing 2 × ChamQ SYBR Color qPCR Master Mix (Nanjing Nuoweizan Biotechnology Co., Ltd., China), 5 µM of each primer, and 2 µL of soil DNA template was mixed. A 10-fold serial dilution of positive plasmid DNA was generated to establish a standard curve. The number of copies of *cbbL* in each sample was calculated from the $C_T$ (threshold cycle) value compared with the standard curve.

## Statistical analysis

Data (means ± SE or SD, $n$ = 3) were analyzed by one-way analysis of variance using IBM SPSS Statistics software v27 (IBM Corp., Armonk, NY, USA), and significant differences between treatments were compared using the Duncan test at $P < 0.05$ (27). Ace and Shannon indices were analyzed using Student's $t$-test in QIIME (v1.9.1) (http://qiime.org/index-qiime1.html). "Stats" for multigroup comparisons was used to

screen significant differences in bacterial species between groups. Redundancy analysis (RDA) was used to evaluate the relationships between SOC, MBC, *cbbL* copies, and RuBisCO activity, and bacterial community structure in soil aggregates under different agronomic measures. The subset of environmental variables with significant correlations with bacterial communities was further identified using the "envfit" function. RDA, principal coordinate analysis (PCoA), and permutational multivariate analysis of variance (PERMANOVA) were performed using the packages "vegan" and "corrplot" in R (28). We used structural equation modeling (SEM) to measure the direct and indirect effects of aggregate sizes and agronomic measures on soil bacterial community diversity, soil carbon components, *cbbL* abundance, and RuBisCO activity in the R (v4.3.1) packages "piecewiseSEM" and "ggplot2."

## RESULTS

### SOC and MBC in different soil aggregate fractions

SOC content in the <0.25 mm fraction, at 9.59–15.15 g kg$^{-1}$, was lower than that in the other aggregate fractions by 13.01% to 34.39%. Compared to the no-fertilizer (Ctrl) and chemical fertilizer (NPK) treatments, the straw plus chemical fertilizer (NPKS) treatment significantly increased SOC content by 32.92%–52.93% and 19.90%–83.81%, respectively ($P < 0.05$) (Fig. 1a). The highest SOC content in the NPKS treatment was recorded in the 0.25–1 mm fraction, at 20.34 g kg$^{-1}$.

Relative to the Ctrl and NPK treatments, the NPKS treatment increased MBC content in various aggregate fractions considerably ($P < 0.05$), especially in the 0.25–1 mm fraction, with increases of 37.18% and 112.03%, respectively (Fig. 1b). Relative to that in the control treatment, MBC content in the >2 mm aggregate in the NPK treatment increased significantly, by 19.81%, but decreased in the 0.25–1 mm fraction, by 35.30% ($P < 0.05$). The average MBC contents of microaggregates (<0.25 mm) were lower than those of the other aggregate fractions in all treatments.

Overall, SOC and MBC contents had strong responses to agronomic measures and soil aggregate separation. The highest SOC and MBC contents were in the 0.25–1 mm fraction, and the lowest were in the <0.25 mm fraction. The SOC and MBC contents in various aggregate fractions under the NPKS treatment were significantly higher than those in the other treatments, and the SOC and MBC contents in the >2 mm fraction under the NPK treatment were higher than those in the control treatment.

### RuBisCO activity and *cbbL* gene abundance in soil aggregates

Relative to the control, NPKS treatment significantly increased RuBisCO activity of aggregates across most size fractions, with the exception of >2 mm aggregate; NPK treatment only increased it in the <0.25 mm fraction ( Fig. 2a). The highest RuBisCO activity under the NPKS (and NPK) treatment was observed in the 0.25–1 mm (and <0.25 mm) aggregate fraction, at 172.92 (and 160.33) nmol h$^{-1}$ g$^{-1}$. RuBisCO activity in various soil aggregates was significantly higher under the NPKS treatment than under the other two treatments. Relative to NPK, NPKS enhanced RuBisCO activity significantly in various aggregate fractions, by 7.29%–26.08% ($P < 0.05$).

The highest *cbbL* abundance in all treatments was observed in the 0.25–1 mm fraction, at 5.51 × 10$^{8}$ (Ctrl), 4.58 × 10$^{8}$ (NPK), and 6.27 × 10$^{8}$ (NPKS) copies g$^{-1}$, respectively (Fig. 2b). Relative to NPK, NPKS increased *cbbL* copies in soil aggregate fractions significantly, by 36.72%–124.89%. However, *cbbL* copies in the 0.25–1 mm fraction showed no significant variation among different agronomic management practices.

### Bacterial community composition in soil aggregates

In the soil aggregate samples, the most abundant bacterial phyla under the NPK and NPKS treatments were Proteobacteria (22.69%–34.60%), followed by Chloroflexi, Actinobacteria, and Acidobacteria (Fig. 3). Soil aggregate separation led to changes in bacterial community structure. In detail, under the Ctrl and NPKS treatments, the highest

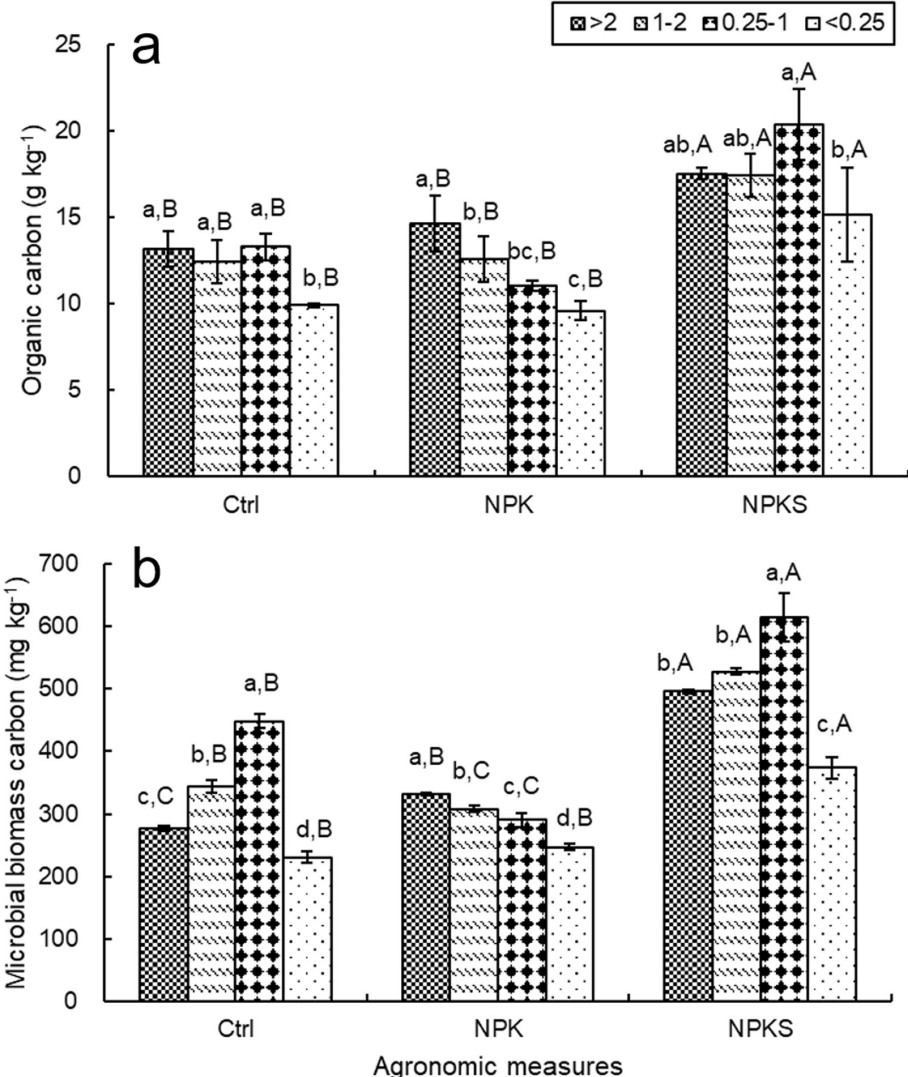

**FIG 1** The contents of organic carbon (a) and microbial biomass carbon (b) in soil aggregates under different management measures. Note: Ctrl, non-fertilizer plot; NPK, pure chemical fertilizer plot; NPKS, chemical fertilizer and straw application plot. Legend (>2, 1–2, 0.25–1, and <1), aggregate size fractions, mm. Different uppercase letters indicated significant difference among different management measures at the same particle size, and different lowercase letters indicated significant difference among different aggregates in the same management measure at the 0.05 level.

relative abundances of Proteobacteria and Actinobacteria were in the 0.25–1 mm aggregate fraction, in which the relative abundances of Chloroflexi and Acidobacteria were the lowest. In the NPK treatment, as the size of the aggregate decreased, the relative abundance of Proteobacteria decreased, whereas the relative abundance of Chloroflexi increased. In addition, the relative abundance of Proteobacteria in soil aggregates rose by approximately 6.85%–29.44% in the NPKS treatment and rose from 21.49%–26.73% to 22.69%–33.52% in the NPK treatment, in comparison with Ctrl. Relative to NPKS, NPK increased the relative abundance of Acidobacteria by 9.23%–80.64%.

Multigroup comparison was performed to compare the relative abundances of bacterial phyla showing significant differences under different agronomic measures and aggregate fractions (Fig. 4). The relative abundances of Gemmatimonadetes, Nitrospirae, and Rokubacteria were significantly higher ($P < 0.001$) in the >1 mm macroaggregate fraction than in the <0.25 mm microaggregate fraction. The relative abundances of

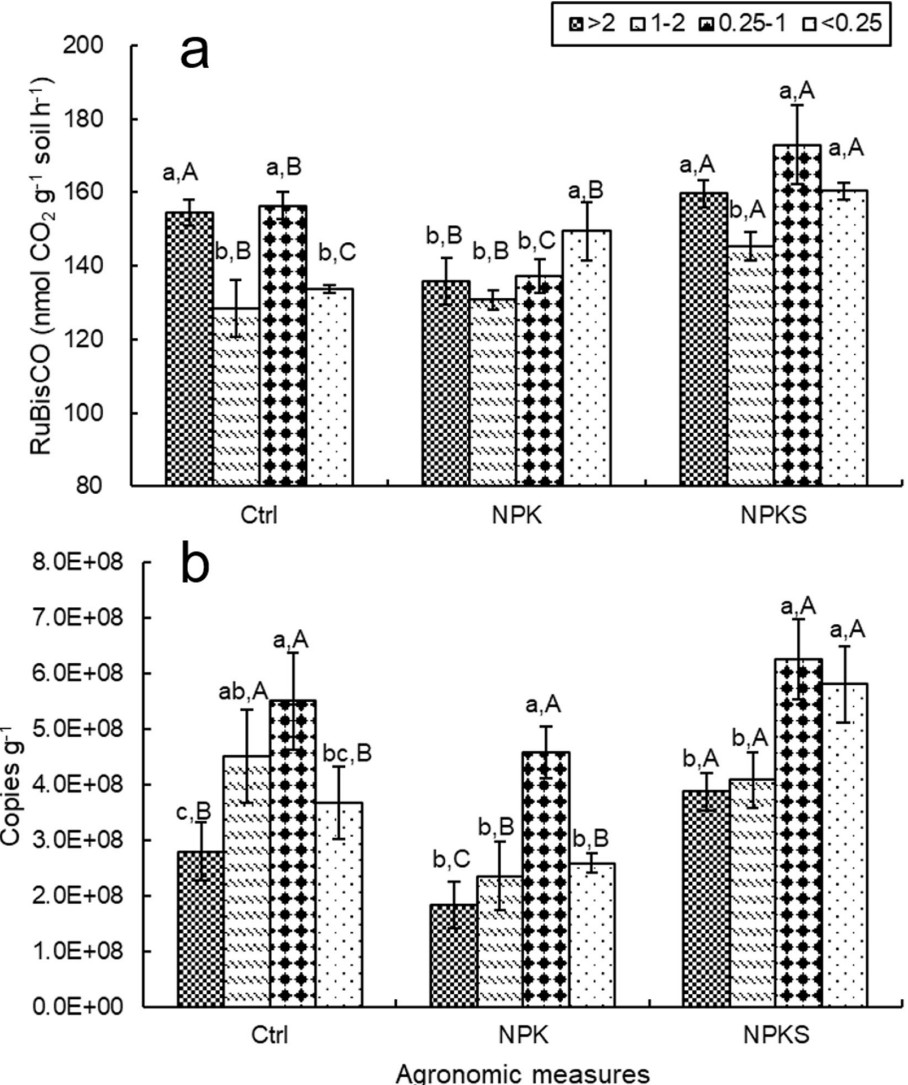

**FIG 2** The activity of RuBisCO (a) and abundance of *cbbL* gene (b) in soil aggregates under different management measures. Note: Ctrl, non-fertilizer plot; NPK, pure chemical fertilizer plot; NPKS, chemical fertilizer and straw application plot. Legend (>2, 1–2, 0.25–1, and <1), aggregate size fractions, mm. Different uppercase letters indicated significant difference among different management measures at the same particle size, and different lowercase letters indicated significant difference among different aggregates in the same management measure at the 0.05 level.

Proteobacteria and Actinobacteria in the aggregates exhibited significant differences ($P < 0.05$) under different agronomic measures.

## Bacterial community diversity in soil aggregates

According to results of PERMANOVA based on the Bray-Curtis algorithm, management measures and soil aggregates altered bacterial community composition greatly ($R^2 = 0.55$, $P = 0.004$), explaining 11% and 18% of the variation in bacterial community structure, respectively (Fig. 5a, Table S1). PCoA showed that different management measures and soil aggregates induced significant changes in bacterial community structure (Fig. 5b), and the first and second axes explained 56.65% and 13.83% of the variation in bacterial community structure, respectively. Bacterial communities in the control and NPK treatments overlapped partially in the second and third quadrants. In particular, <0.25 mm aggregates treated by control and NPK were clustered in the second quadrant.

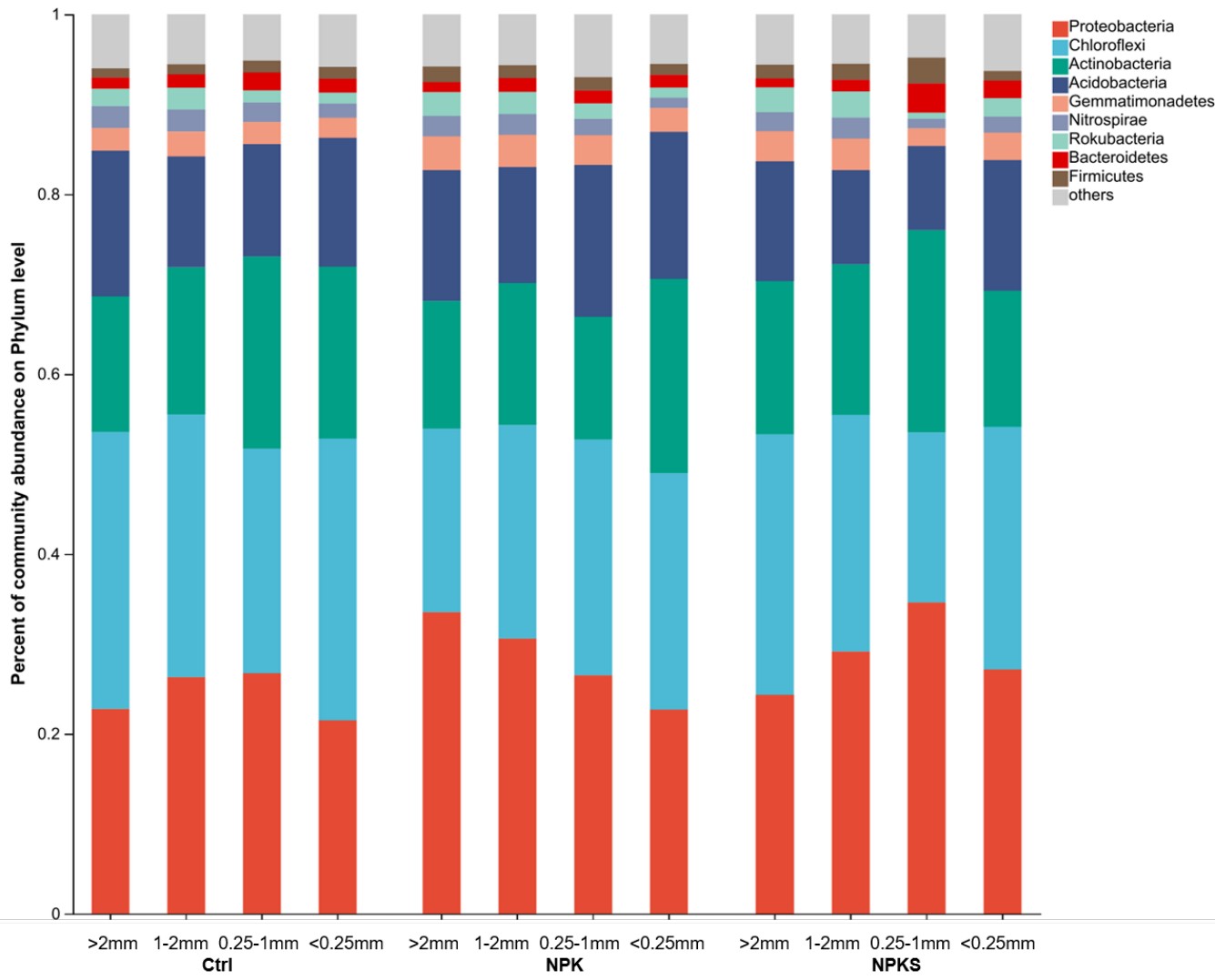

**FIG 3** Relative abundance of the bacterial phyla in soil aggregates under different management measures. Less than 2% abundance of the phyla was merged into others.

The diversity and richness of soil bacterial communities (Shannon and Ace indexes) varied across soil aggregate fractions under three management measures (Fig. 6). Generally, the Shannon and Ace indexes of the <1 mm aggregates were higher compared to >1 mm aggregates. Compared with that of the NPKS treatment, the Shannon index of aggregates in the Ctrl and NPK treatments decreased by approximately 0.44% and 1.04%, respectively. The maximum Ace indexes of the NPK and NPKS treatments were observed in the 0.25–1 mm fraction, as 3993 and 4220, respectively. In addition, the mean Ace index in soil aggregates in the NPKS treatment was significantly higher, by 8.14% ($P < 0.05$), than that in the NPK treatment.

## Relationships between soil carbon and bacterial community

Soil bacterial community composition was closely correlated with soil biochemical properties. The aggregate-associated SOC ($R^2 = 0.23$, $P < 0.05$) and MBC ($R^2 = 0.25$, $P < 0.01$) were closely correlated with bacterial community composition (Fig. S1). For example, SOC and MBC were positively correlated with Proteobacteria and Actinobacteria, and negatively correlated with Chloroflexi and Acidobacteria.

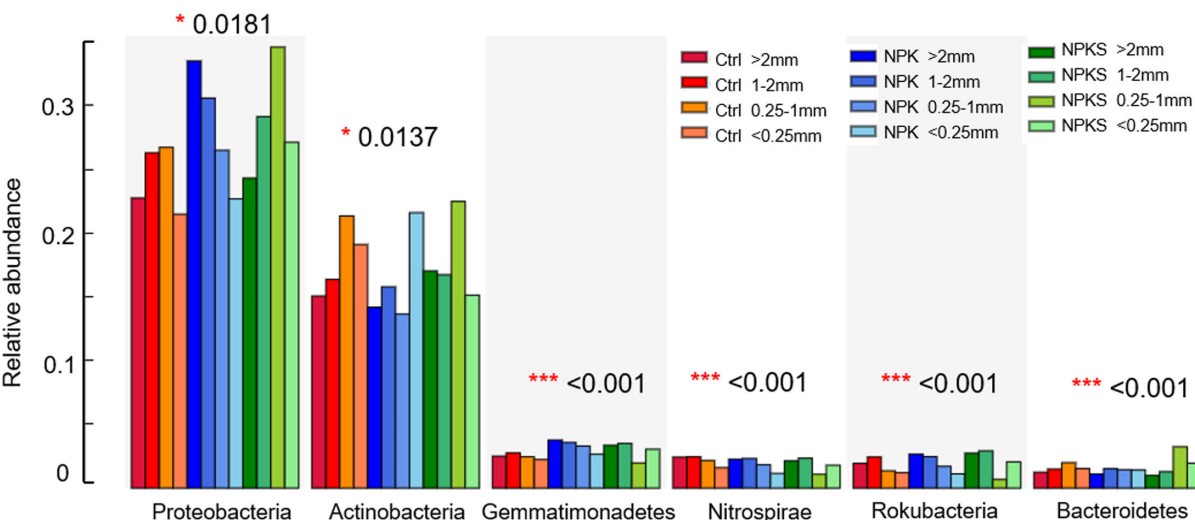

**FIG 4** Relative abundance of bacterial phyla showing significant differences in soil aggregates under different management measures (only the top six with significant differences are shown). A one-way ANOVA was used to evaluate the significance of differences between the indicated groups. *$P < 0.05$; **$P < 0.01$; ***$P < 0.001$.

In detail, the cumulative variance explained by RDA1 and RDA2 was 85.75%, 84.08%, 86.44%, and 65.42% for the four soil aggregates fractions (<0.25 mm, 0.25-1 mm, 1–2 mm, and >2 mm), respectively (Fig. 7). RuBisCO activity ($R^2 = 0.85$, $P < 0.01$) was strongly correlated with bacterial community composition in the >2 mm aggregate, similar to *cbbL* gene copy number ($R^2 = 0.74$, $P < 0.05$) in the 1–2 mm aggregate. In the <0.25 mm aggregate, SOC ($R^2 = 0.87$, $P < 0.001$), MBC ($R^2 = 0.97$, $P < 0.001$), *cbbL* gene copies ($R^2 = 0.82$, $P < 0.05$), and RuBisCO activity ($R^2 = 0.69$, $P < 0.05$) responded strongly to bacterial community composition. RDA also showed that the dominant bacteria, Proteobacteria, were negatively correlated with soil biochemical properties in the >2 mm aggregate fraction (and *cbbL* copies in the 1–2 mm aggregate), but positively correlated with soil biochemical properties in the <1 mm aggregate fraction. Chloroflexi exhibited trends almost opposite to those of Proteobacteria, separated by RDA1.

A structural equation model (SEM) was used to quantify the impact of long-term agronomic measures and aggregate sizes on soil microbial activity and SOC content (Fig. 8). According to the results, agronomic measures and aggregate sizes directly or indirectly explained 83% of SOC variation and 79% of MBC variation, respectively, by influencing soil biochemical properties and bacterial attributes. However, in soil aggregates, MBC content was directly negatively influenced by bacterial community (path coefficient = −0.53, $P < 0.001$), which promoted RuBisCO activity by increasing *cbbL* abundance (path coefficient = 0.67, $P < 0.001$) and finally positively affected SOC content.

## DISCUSSION

### Effects of straw application on carbon composition and soil aggregate biological activity

Straw application and fertilization affected SOC and MBC distribution across soil aggregate size fractions (Fig. 1), indicating that agronomic measures influence the soil carbon occurrence mechanism in aggregates. The NPKS treatment increased SOC and MBC contents in soil aggregates considerably compared to other treatments (Fig. 1). On the one hand, straw input as a carbon source could directly increase SOC content. On the other hand, straw promoted aggregate formation, which effectively prevented SOC within the aggregates from being decomposed preferentially, so that it increased SOC accumulation in aggregates indirectly (29). Also, straw return accelerates soil aggregates'

biological activities that convert the straw carbon into MBC (30). The above results indicate that reasonable field management is beneficial to SOC content and soil quality enhancement.

In rice–wheat rotation soil aggregates, the results showed that three management measures significantly influenced RuBisCO activity and *cbbL* abundance. Notably, the straw return plus fertilizer (NPKS) treatment had the strongest positive effect (Fig. 2). Straw and appropriate quick-acting fertilizer are potential substrates and sources for carbon fixation by soil microorganisms. The results support the perspective that a rise in soil nutrients could promote soil RuBisCO activity and *cbbL* abundance (31). On the contrary, long-term NPK and Ctrl treatments decreased MBC contents and RuBisCO activity in aggregates, resulting in a relatively low carbon turnover, which may reduce soil carbon assimilation capacity and destroy soil aggregate structure (32). The findings offer crucial insights into the aggregate carbon changes caused by agronomic management measures, which could be exploited to enhance straw resource use and soil carbon sustainability.

The response of bacterial diversity (Shannon and Ace indices) to agronomic management measures varied in different soil aggregate fractions (Fig. 6). In our study, relative to the control treatment, long-term chemical fertilizer application (NPK) decreased bacterial community diversity in aggregate fractions, which were stabilized and restored with straw return plus chemical fertilizer (NPKS treatment). Similarly, according to Wu et al. (33), organic agricultural material input could mitigate the detrimental impacts of reduced bacterial diversity due to nutrient or energy deficiencies following chemical fertilizer input.

In rice–wheat rotation soil, Proteobacteria, Chloroflexi, Actinobacteria, and Acidobacteria were the predominant bacterial groups (Fig. 3), which is consistent with Wang et al. (15), indicating that these bacteria adapt easily to various habitats and evolved into dominant communities (23). Chloroflexi and Acidobacteria are slow-growing oligotrophic taxa (k-strategists) (34), whereas Proteobacteria and Actinobacteria are fast-growing copiotrophic taxa (r-strategists) (35). Accordingly, the high abundance of Proteobacteria in the NPKS treatment indicated that straw return plus fertilizer promoted active and nutrient-rich microbial communities by supporting a more fertile soil environment. In our previous work, the NPK treatment aggravated soil acidification (25), which may favor the growth and reproduction of Acidobacteria, relative to straw return combined with chemical fertilizer. As a result, long-term application of pure chemical fertilizer decreased SOC and MBC contents in the <2 mm aggregate fraction (Fig. 1).

Previous studies have reported higher carbon content in macroaggregates than in microaggregates (29, 36). Similarly, in this work, the 0.25–1 mm aggregate fraction was dominated by macroaggregates and was relatively sensitive to straw return (Fig. 1). In addition, rice–wheat rotation soil aggregates were in periodic dry–wet alternation, with the 0.25–2 mm aggregates being dominant. Consequently, aggregate size plays an important role in organic carbon transformation and accumulation (37). As shown in Fig. 1, greater SOC and MBC accumulation were observed in the >0.25 mm aggregates than in the <0.25 mm aggregates, which is consistent with Wang et al. (38) and Zheng et al. (23). According to the Aggregate Hierarchy Conceptual Model (39), macroaggregates are composed primarily of microaggregates, microorganisms, fungal hyphae, and organic binding agents, resulting in richer organic carbon components in macroaggregates than in microaggregates. Our previous studies have also observed significantly higher levels of water-stable aggregates in the >0.25 mm fraction compared with the <0.25 mm fraction, which may also be a key reason for the high SOC and MBC contents in macroaggregates.

## Influence of soil aggregate size on microbial regulation of soil organic carbon storage

Same with Zheng et al. (23), who conducted research in an orchard and reported that increasing soil aggregate size led to a decrease in bacterial diversity and richness

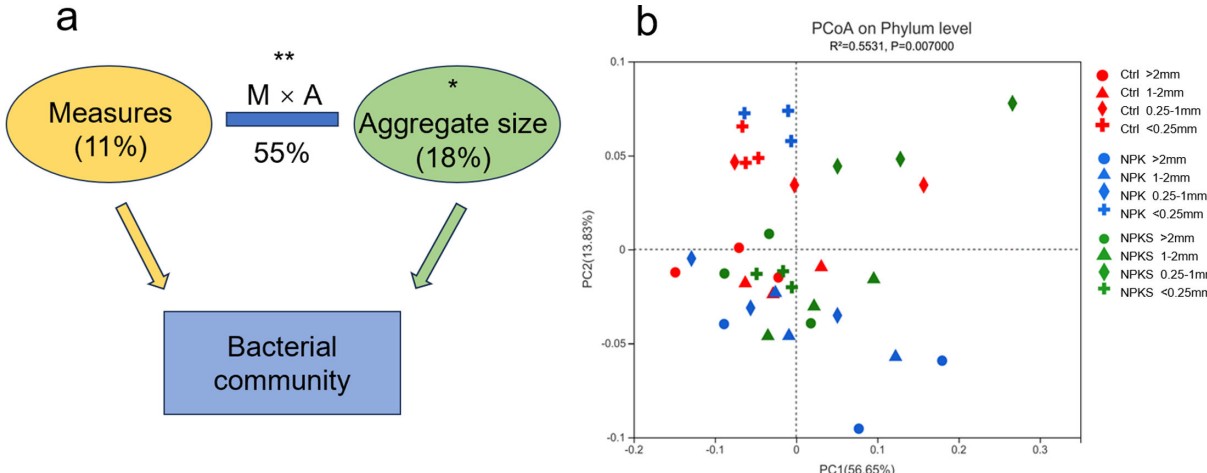

**FIG 5** Permutational multivariate analysis of variance (PERMANOVA, a) and principal coordinate analysis (PCoA, b) of the bacterial community in soil aggregates under different management measures. "M" means agronomic measures, "A" means aggregate size. Distances between symbols on the ordination plot reflect relative dissimilarities in community structures.

indices. We also found that the bacterial community was directly negatively influenced by aggregate size in the SEM model (Fig. 8). The discrepancy could be attributed to long-term differences in farmland use type as well as agronomic practices (irrigation, fertilization, and so on) (40). Tiny soil pores between aggregates could shield bacteria from soil fauna, so that <1 mm aggregates provide soil bacteria with a stable habitat and promote bacterial diversity (41). Unexpectedly, MBC content showed a negative correlation with bacterial community diversity (Fig. 8). This pattern likely reflects the dominant role of microbial life-history strategy selection (k/r-strategists) in aggregate (42), suggesting decreased microbial carbon use efficiency or the dominance of bacterial taxa contributing less to MBC. Accordingly, Proteobacteria and Actinobacteria were enriched in the 0.25–1 mm macroaggregates of the NPKS treatment compared with the other soil samples, which promoted SOC accumulation in soil in the 0.25–1 mm macroaggregates.

RDA analysis confirmed that SOC and MBC were strongly correlated with bacterial community structure in soil aggregates (Fig. S1). The porosity, nutrient contents, and carbon availability of aggregates with suitable particle size were higher than those of larger or smaller aggregates (43). Therefore, the contents of SOC and MBC driven by microbial activity in the 0.25–1 mm aggregate fraction were higher than those in the <0.25 mm and >2 mm fractions. Conversely, relative to straw application, there is low substrate supply in the poor soil structure following long-term NPK application (25), which is consistent with our study. This explains the decreased Proteobacteria and increased Chloroflex in the NPK treatment, with a decrease in soil aggregate size.

With suitable porosity and high nutrient availability, the 0.25–1 mm aggregate fraction was the main soil microorganism habitat and promoted SOC cycling (44). However, in the >2 mm and 0.25–1 mm aggregate fractions, NPK reduced soil RuBisCO activity and *cbbL* abundance relative to those in the control treatment (Fig. 2). $CO_2$ assimilation capacity might have been impaired by decreased soil fertility and activity following pure NPK application for 35 years. Zhou et al. (21) and Guo et al. (45) also reported that, although chemical fertilization could enhance soil available nutrients, prolonged application could lead to soil acidification or structural degradation, potentially diminishing RuBisCO activity and *cbbL* abundance in soil aggregates.

In most agricultural soils, *cbbL*-carrying facultative bacteria are predominant (17), whereas gene presence does not necessarily translate to functional expression. $CO_2$ assimilation (the Calvin cycle with a key enzyme RuBisCO) is an alternative pathway for facultative autotrophs to synthesize their cellular components when organic carbon

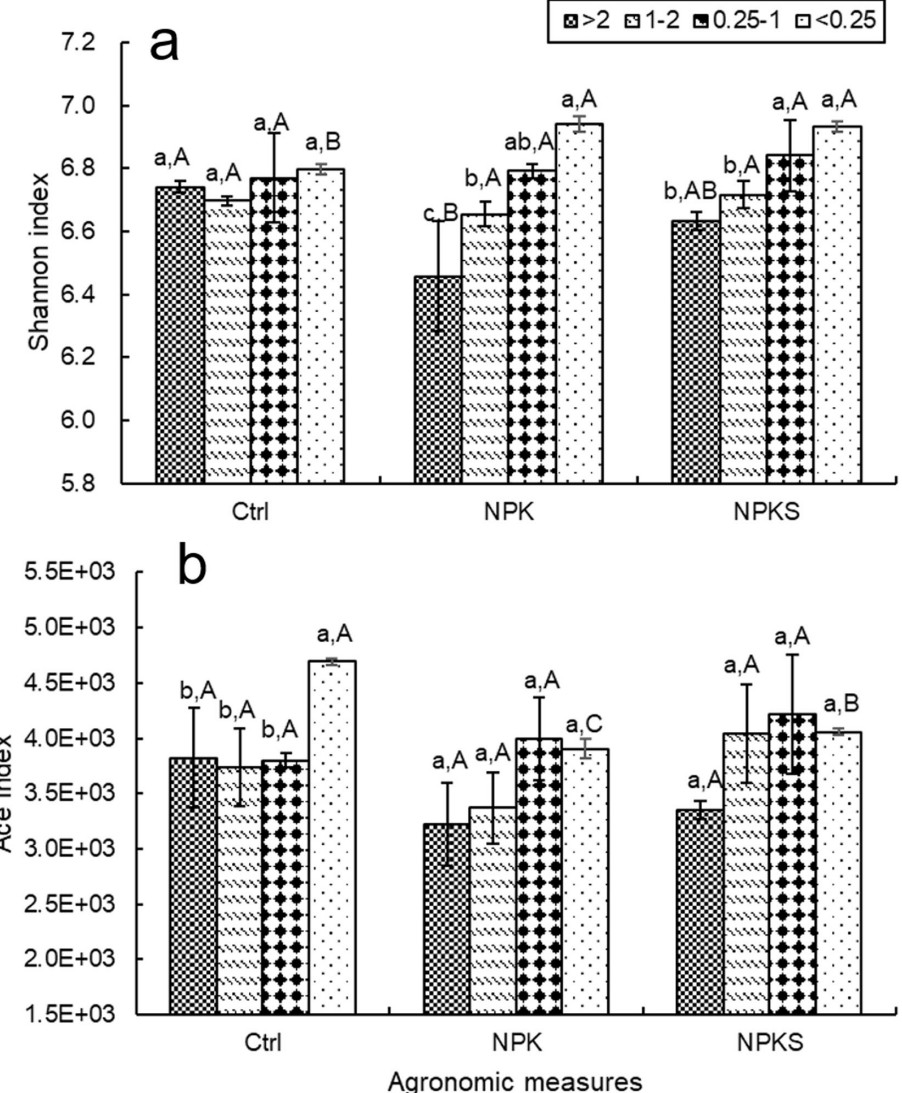

**FIG 6** The α diversity (Shannon [a] and Ace [b] indices) of bacteria in soil aggregates under different management measures. The ordinate is the exponential average of each group. Note: Ctrl, non-fertilizer plot; NPK, pure chemical fertilizer plot; NPKS, chemical fertilizer and straw application plot. Legend (>2, 1–2, 0.25–1, and <1), aggregate size fractions, mm. Different uppercase letters indicated significant difference among different management measures at the same particle size, and different lowercase letters indicated significant difference among different aggregates in the same management measure at the 0.05 level.

sources are unavailable (e.g., NPK treatment). Moreover, in our qPCR amplification, only *cbbL* (I) was detected, which might lead to differences in RuBisCO activity and *cbbL* abundance (26). Thus, when RuBisCO activity was inconsistent with changes in *cbbL* gene abundance in NPK-treated <0.25 mm aggregate (Fig. 2), we propose that RuBisCO activity better reflects the realized autotrophic carbon sequestration capacity in soil aggregates, relative to *cbbL* abundance. We also found that the functional gene *cbbL* is more sensitive to the bacterial community, which, in turn, has consequences for RuBisCO activity directly linked to SOC. As reported by Trivedi et al. (46), the results indicate that the soil microbial community regulates enzyme activity through functional gene abundance. Therefore, Rubisco activity may be a better predictor for overall carbon fixation dynamics (fixation, release, storage) than the abundance of *cbbL* in the community. In our work, limited progress has been made in promoting soil carbon

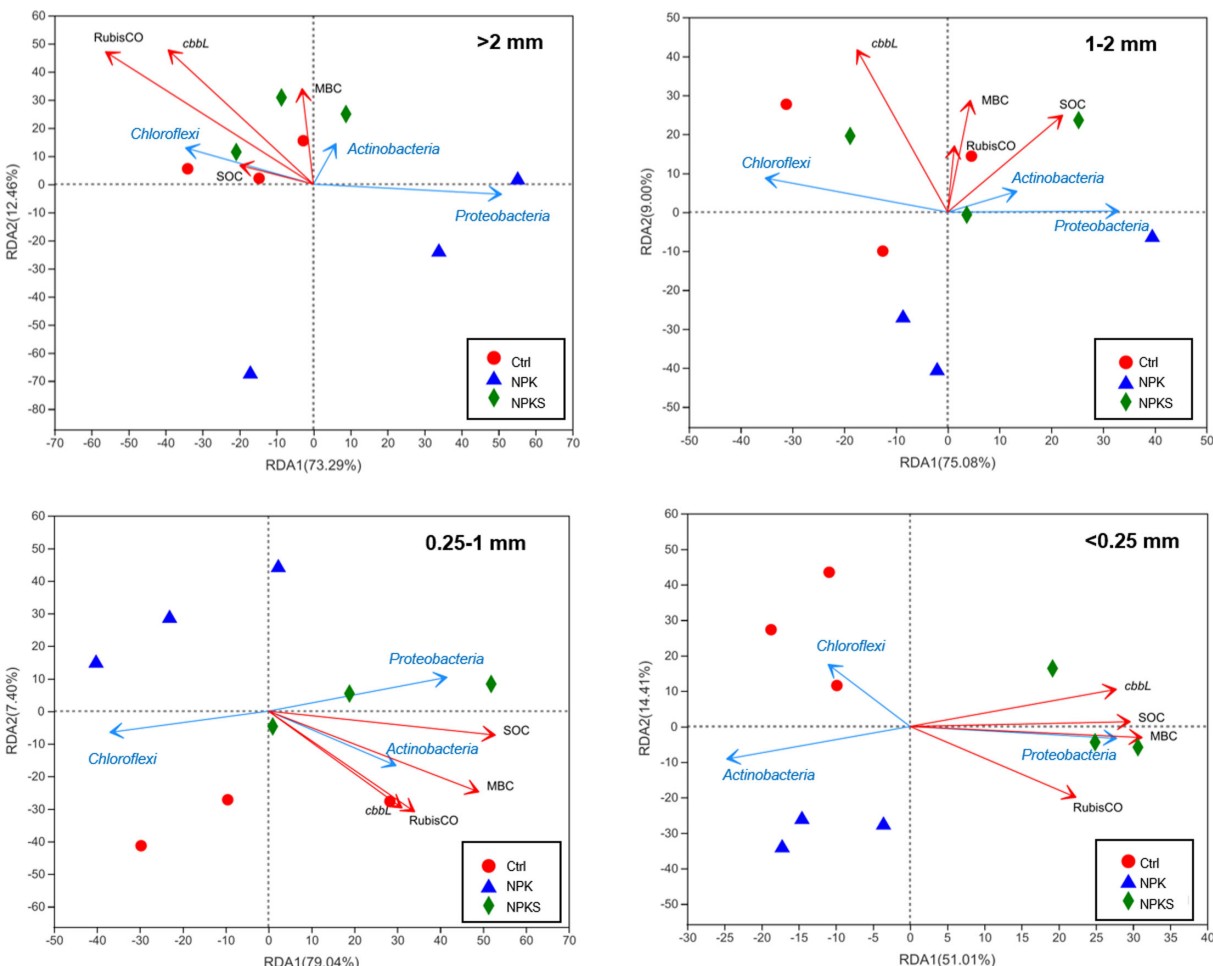

**FIG 7** RDA showing the influence of soil properties (SOC, MBC, cbbL gene copies, and RuBisCO activity) on bacterial community (phylum level) in different soil aggregates under three management measures. The top three dominant bacterial phyla are shown.

storage. Future metagenomic efforts will explore more RuBisCO forms, particularly Form II RuBisCO, which has been found to operate twice as fast as the fastest carboxylation rate measured RuBisCO to date (47). The potential implications of Form II RuBisCO for carbon fixation have likely been underestimated.

## Conclusions

Soil carbon and bacterial community dynamics in aggregates are influenced by long-term agronomic management measures. In straw return plus chemical fertilizer treatment, SOC and MBC contents in soil aggregates were closely correlated with the bacterial community. Straw amendment affects the bacterial community structure of aggregates and favors the increase in SOC content. Based on our model, remodeling of bacterial community structure in the 0.25–1 mm aggregate fraction increases *cbbL* abundance, which leads to increased Rubisco activity in the community. This, in turn, increases SOC and MBC content. RuBisCO activity, *cbbL* abundance, and bacterial community diversity were relatively high in the 0.25–1 mm aggregate fraction, which provided a good microenvironment for soil carbon renewal and transformation.

## ACKNOWLEDGMENTS

This work was supported by the National Natural Science Foundation of China (42377333); the National Key Research and Development Program of China

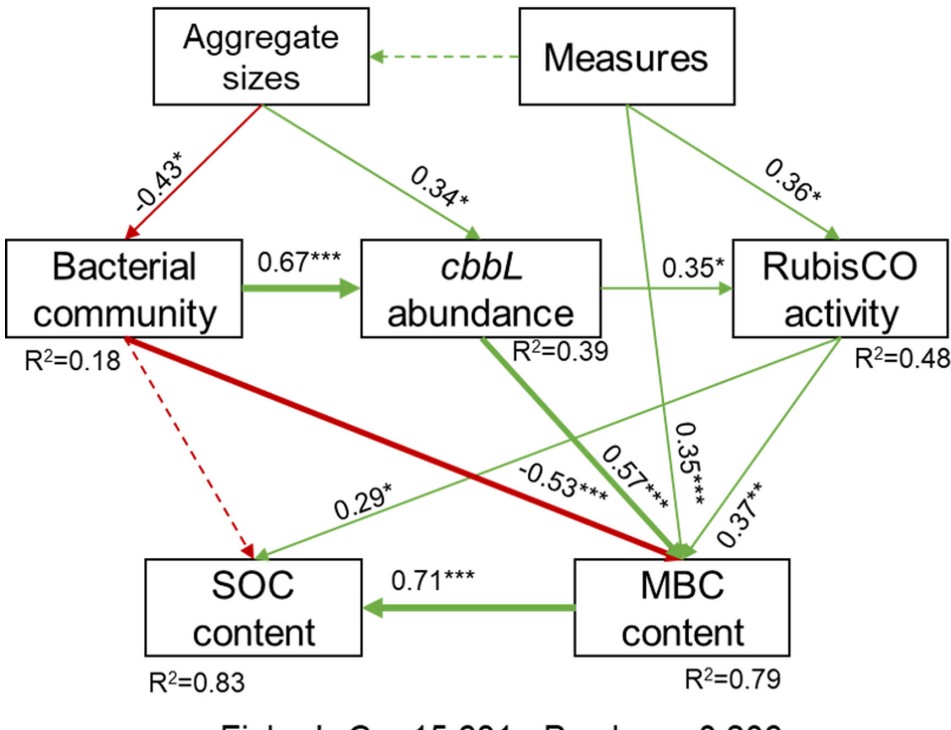

**FIG 8** SEM showing the effects of agronomic measures and aggregate size on SOC and MBC content. Note: the green and red arrows represent the positive and negative impact, respectively. Solid and dashed arrows suggested the significant ($P < 0.05$) and non-significant effects ($P > 0.05$), respectively. Significant level: *$P < 0.05$, **$P < 0.01$, and ***$P < 0.001$. The thickness of the arrow and the value next to the arrow represent the size of the path coefficient. $R^2$ alongside square boxes indicated that the variable was explained by proportion.

(2023YFD1901202); and the Key Program of China National Tobacco Corporation Sichuan (SCYC202206).

The authors confirm contribution to the paper as follows: X.L., conceptualization, data curation, formal analysis, investigation, writing—original draft, and writing—review and editing; R.H., conceptualization, data curation, supervision, visualization, and writing—review and editing; Y.W., funding acquisition, investigation, methodology, and resources; H.J., funding acquisition and resources; Y.L., data curation, formal analysis, and investigation; C.W., funding acquisition, project administration, and supervision; B.L., conceptualization, funding acquisition, resources, project administration, supervision, and writing—review and editing.

## AUTHOR AFFILIATIONS

[1]College of Resource, Sichuan Agricultural University, Chengdu, China
[2]Liangshan Branch of Sichuan Provincial Tobacco Company, Xichang, China
[3]China National Tobacco Corporation Sichuan, Chengdu, China

## AUTHOR ORCIDs

Bing Li  http://orcid.org/0000-0001-9625-3181

## FUNDING

| Funder | Grant(s) | Author(s) |
| --- | --- | --- |
| National Natural Science Foundation of China | 42377333 | Bing Li |

| Funder | Grant(s) | Author(s) |
|---|---|---|
| National Key Research and Development Program of China | 2023YFD1901202 | Bing Li |
| Key program of China national tobacco corporation Sichuan | SCYC202206 | Yong Wang |

## AUTHOR CONTRIBUTIONS

Xinyue Li, Conceptualization, Data curation, Formal analysis, Investigation, Writing – original draft, Writing – review and editing | Rong Huang, Conceptualization, Data curation, Supervision, visualization, Writing – review and editing | Yong Wang, Funding acquisition, Investigation, Methodology, Resources | Hong Jiang, Funding acquisition, Resources | Youlin Luo, Data curation, Formal analysis, Investigation | Changquan Wang, Funding acquisition, Project administration, Supervision | Bing Li, Conceptualization, Funding acquisition, Project administration, Resources, Supervision, Writing – review and editing

## ADDITIONAL FILES

The following material is available online.

### Supplemental Material

**Supplemental material (Spectrum00088-25-s0001.docx).** Table S1; Fig. S1

### Open Peer Review

**PEER REVIEW HISTORY (review-history.pdf).** An accounting of the reviewer comments and feedback.

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
