## [Reviewer comments · Microbiology Spectrum]

Microbiology Spectrum

Straw application promotes soil carbon storage by affecting aggregate-associated bacterial community structure and RuBisCO activity: A 35-year field experiment

Xinyue Li, Rong Huang, Yong Wang, Hong Jiang, Youlin Luo, Changquan Wang, and Bing Li

Corresponding Author(s): Bing Li, Sichuan Agricultural University - Chengdu Campus

Review Timeline:

Submission Date:	January 17, 2025
Editorial Decision:	March 21, 2025
Revision Received:	April 22, 2025
Editorial Decision:	May 14, 2025
Revision Received:	May 26, 2025
Editorial Decision:	June 9, 2025
Revision Received:	June 11, 2025
Accepted:	June 19, 2025

Editor: Weimin Sun

Reviewer(s): Disclosure of reviewer identity is with reference to reviewer comments included in decision letter(s). The following individuals involved in review of your submission have agreed to reveal their identity: Yudan Bai (Reviewer #3)

Transaction Report:

DOI: <https://doi.org/10.1128/spectrum.00088-25>

Re: Spectrum00088-25 (Straw application promotes soil carbon storage by regulating aggregate-associated bacterial community structure and RuBisCO activity: A 35-year field experiment)

Dear Dr. Bing Li:

Thank you for the privilege of reviewing your work. Below you will find my comments, instructions from the Spectrum editorial office, and the reviewer comments.

Both reviewers suggested the writing and formatting of the manuscript needs substantial improvement, especially the misleading terminology. The authors may pay more attention to writing in the revised manuscript.

Revision Guidelines

Sincerely,
Weimin Sun
Editor
Microbiology Spectrum

Reviewer #1 (Comments for the Author):

This paper addresses how soil organic and microbial biomass carbon is affected by different agronomic management practices, and concludes that soil organic and microbial biomass carbon is increased by using a method that supplements chemical fertilizers with straw.

The language used throughout the paper was sometimes misleading or confusing. Especially in the conclusion section, terminology was used that was inappropriate for the measured values and work presented. In particular:

- A) "autotrophic carbon fixation capacity" in lines 387-389. I did not see any results that looked at total carbon fixation rates in these samples. If you are using something else as a proxy, please indicate what you are using.
- B) "bacterial activity" in lines 403-404: Bacterial activity was not reported in this study (usually measured as respiration through $3H$ consumption). Are you referring to Rubisco activity? Or MBC?
- C) "regulation" throughout the paper to mean a causal relationship rather than a specific term used in gene expression and enzyme activity. Regulation should not be used in these contexts. Something like "causes", "is correlated with", "increases", "decreases", any of these would be fine, but "regulates" is not appropriate.

In addition, some figures were difficult to understand. Specifically, the ways to indicate significance were often extremely difficult to follow and I would consider other ways that they could be depicted. Some specific comments about the figures follow:

- A) Fig. 8 colors should be switched. Red is typically associated with negative while green is typically associated with positive. Either that, or use an arrowhead like \rightarrow for positive and like $\left|$ for negative, as are often used with gene interaction maps.
- B) The change in labels from size fractions to a, b, c, d in Figs. 3-6 is quite confusing. I would suggest continuing to label with the size fraction numbers.
- C) Significance indicators on Figs. 1, 2 and 6 were difficult to read and understand. First, I would use commas to separate the significance between the different aggregates from the significance between the treatments, e.g. "ab,A" just to separate the two more. Second, the Shannon/Ace index significance indicators are exceedingly difficult to read. Is there another way that you can represent the significance in Fig. 6?
- D) If you begin by clustering based on size fraction, I would continue to cluster by size fraction rather than switching to clustering by treatment halfway through.
- E) Fig. 5: What does M x A represent? The measures & aggregate size together?

I also have some methodological questions:

- 1) Can you comment on whether the qPCR primers are able to detect form II cbbL, or just form I? Given the high percentage of proteobacteria in these studies, it could be relevant to interpreting the results. If these primers are not able to detect form II cbbL, that should be noted in the discussion of the qPCR results as it will mean the Rubisco activity measurements and Rubisco abundance from qPCR are not measuring the same pool of Rubisco.
- 2) The Rubisco activity units need more explanation. Is it $\text{nmol CO}_2 \cdot \text{h}^{-1} \cdot \text{g Rbc}^{-1}$? Or is the grams section referring to the soil solution? Activity is typically reported as $\mu\text{mol CO}_2 \cdot \text{min}^{-1} \cdot \text{mg Rbc}^{-1}$ or $\mu\text{mol CO}_2 \cdot \text{min}^{-1} \cdot \text{mol Rbc active site}^{-1}$.
- 3) It is unclear if you quantified Rubisco content of the soil extracts, and you did not quantify how many active sites were functional, so you did not measure if the enzyme was getting more active. Likely, the "Rubisco activity" results represent an increase in the amount of active Rubisco present in the community, which is a different thing than increasing activity of the protein itself. I would like to see you mention this in your discussion to clarify the interpretation of the results, as the latter is far more likely than the former.
- 4) I am not familiar with your structural equation modeling, and there were very few details in the Methods on how it was done, other than which R package was used - what settings did you use?

Line-by-line Comments:

- 1) Line 12: Abbreviating carbon to C is unnecessarily confusing when you are already going to abbreviate soil organic carbon and microbial biomass carbon to SOC and MBC. I would recommend using "carbon" when you are not using one of the other defined acronyms.
- 2) Line 69-70: What is the relationship between SOC transformation, microbial communities, and enzyme activity? I'd prefer to see the results of the study written here instead of an indication that it has been studied.
- 3) Line 89: Would, or does? If this is not a hypothesis, "Microbial growth and function varies..." is more appropriate.
- 4) Line 149: This method is typically called the "NADH-coupled spectrophotometric" method, as it does not involve fluorescence. Also, please cite the Wu et al. 2014 study in addition to the currently cited study, as that is the one where the method for assessing the Rubisco activity is described in detail.
- 5) Lines 366-369: What exactly are you referring to when you say nutrient and energy supply? Is it an assumption based on the previous study cited in this paragraph, or is it based on specific measured values in this study? If the former, it would be helpful if you reworded it to indicate that your results are consistent with the previous study instead of combining them into one sentence with no way to distinguish where the information is coming from.
- 6) Lines 390-392: I do not understand what this sentence is indicating, as it seems to contradict the sentence before it. Is the better predictor for overall carbon fixation dynamics cbbL or Rubisco activity? If the latter, I would suggest something like, "Thus Rubisco activity may be a better predictor for overall carbon fixation dynamics (fixation, release, storage) than the abundance of cbbL in the community."
- 7) I don't think section 3.2 currently depicts the Rubisco activity results in enough detail. Lines 224-225 state that Rubisco activity was increased significantly in the NPKS in the 0.25-1-mm aggregate class, but it also increased significantly in the <0.25

and 1-2. I would suggest: "...was increased significantly in most size fractions, with the exception of the >2mm aggregates (Fig. 2a)." I also would like to see you discuss that the NPK treatment decreased Rubisco activity in a couple of the size fractions.

8) Lines 406-407: I don't think you have the support for this sentence as it is written. As stated above, regulation is not an appropriate word to use in this context. I would suggest something more like "Based on our model, remodeling of bacterial community structure in the .25-1-mm aggregate fraction increases cbbL abundance, which leads to increased Rubisco activity in the community. This in turn increases SOC..."

Reviewer #3 (Comments for the Author):

This study exposes that straw application promotes soil carbon storage by regulating-associated bacterial community structure and RuBisCO activity. The purpose of the study is clear. However, the manuscript still has more problems in writing and basic formatting as follows:

1. Line 23: "Bacterial community richness and diversity....." is redundant because bacterial community diversity includes richness. Check and modify the whole manuscript.
2. Line 96-100: The authors mentioned in hypothesis (2) that: the 0.25-1-mm aggregate fraction is the main site of C fixation, with greater microbial biomass, cbbL abundance, and RuBisCO activity, promoting C renewal. However, the introductory section only mentions the advantages of macroaggregates (>0.25 mm size) and does not point out the specificity of the size 0.25-1 mm, so such an assumption is unfounded.
3. Use spaces rather than hyphens between values and units, e.g. 2-mm to 2 mm. Check and modify the whole manuscript.
4. Line 389: "As reported by Trivedi et al. (45), the results of the present study indicate that....", this sentence miscellaneous.
5. The manuscript has too many sentences like "in the present study" that are redundant, suggest deleting them and just writing the results. Check and revise the entire manuscript.
6. Line 374-292: Is it better to change the bacterial community structure to bacterial community composition in part 3.5?
7. Line 393-400: There is no discussion in this passage, but rather a presentation of the results of an SEM model, which is not quite appropriate for the author to place in the discussion section.
8. The size of the font needs to be uniform across all figures. The use of black should be avoided in the colours of Figures_1 and Figures_2. Figure_4 lacks vertical labels. The statistical analyses in Figure_6 are better changed to multiple comparisons with one-way ANOVA, i.e., the differences between groups are replaced by a, b, c, etc., for visual clarity.
9. Manuscripts need to be in Times New Roman font. Attention should be paid to basic formatting.

Comments on the review

This study exposes that straw application promotes soil carbon storage by regulating-associated bacterial community structure and RuBisCO activity. The purpose of the study is clear. However, the manuscript still has more problems in writing and basic formatting as follows:

1. Line 23: “Bacterial community richness and diversity.....” is redundant because bacterial community diversity includes richness. Check and modify the whole manuscript.
2. Line 96-100: The authors mentioned in hypothesis (2) that: the 0.25—1-mm aggregate fraction is the main site of C fixation, with greater microbial biomass, *cbbL* abundance, and RuBisCO activity, promoting C renewal. However, the introductory section only mentions the advantages of macroaggregates (>0.25 mm size) and does not point out the specificity of the size 0.25—1 mm, so such an assumption is unfounded.
3. Use spaces rather than hyphens between values and units, e.g. 2-mm to 2 mm. Check and modify the whole manuscript.
4. Line 389: “As reported by Trivedi et al. (45), the results of the present study indicate that....”, this sentence miscellaneous.
5. The manuscript has too many sentences like “in the present study” that are redundant, suggest deleting them and just writing the results. Check and revise the entire manuscript.
6. Line 374-292: Is it better to change the bacterial community structure to bacterial community composition in part 3.5?
7. Line 393-400: There is no discussion in this passage, but rather a presentation of the results of an SEM model, which is not quite appropriate for the author to place in the discussion section.
8. The size of the font needs to be uniform across all figures. The use of black should be avoided in the colours of Figures_1 and Figures_2. Figure_4 lacks vertical labels.

The statistical analyses in Figure_6 are better changed to multiple comparisons with one-way ANOVA, i.e., the differences between groups are replaced by a, b, c, etc., for visual clarity.

9. Manuscripts need to be in Times New Roman font. Attention should be paid to basic formatting.

Response to Reviews

We would like to thank the editors and the reviewers for their insightful and constructive comments on our manuscript. We have carefully considered all the comments, and revised the manuscript accordingly. The English writing was polished by the Transystem (<http://www.transystem.com/>).

Below we provide a point-by-point response to each of the comments raised by the reviewers. The reviewers' comments are written in bold and our responses are in normal characters. The page and line numbers in our response correspond to those in the revised manuscript.

Reviewer #1 (Comments for the Author):

This paper addresses how soil organic and microbial biomass carbon is affected by different agronomic management practices, and concludes that soil organic and microbial biomass carbon is increased by using a method that supplements chemical fertilizers with straw. The language used throughout the paper was sometimes misleading or confusing. Especially in the conclusion section, terminology was used that was inappropriate for the measured values and work presented. In particular:

A) "autotrophic carbon fixation capacity" in lines 387-389. I did not see any results that looked at total carbon fixation rates in these samples. If you are using something else as a proxy, please indicate what you are using.

Answer: Thanks for your comments. This paragraph has been reorganized in the revised manuscript. (Lines 383-398)

RuBisCO and *cbbL* were considered to indicate autotrophic carbon sequestration capacity (Yuan et al. 2012). We introduced it in the "Introduction" part, the Calvin cycle is the main pathway for CO₂ assimilation in autotrophic microorganisms. We propose that RuBisCO activity serves as a more reliable indicator for autotrophic C sequestration potential in soil aggregates, relative to *cbbL* abundance.

[1] Yuan H, Ge T, Chen C, O'Donnell AG, Wu J. 2012. Significant role for microbial autotrophy

in the sequestration of soil carbon. *Appl Environ Microbiol.* 78, 2328-2336.

B) "bacterial activity" in lines 403-404: Bacterial activity was not reported in this study (usually measured as respiration through 3H consumption). Are you referring to Rubisco activity? Or MBC?

Answer: Thanks for your comments. This sentence has been rewritten to "SOC and MBC contents in soil aggregates were closely correlated with bacterial community." in the revised manuscript. (Lines 402)

C) "regulation" throughout the paper to mean a causal relationship rather than a specific term used in gene expression and enzyme activity. Regulation should not be used in these contexts. Something like "causes", "is correlated with", "increases", "decreases", any of these would be fine, but "regulates" is not appropriate.

Answer: Thanks for your comments. These "regulation" have been replaced by "affect", "increases" and so on. We keep them only in the description of gene expression and enzyme activity. (Lines 1, 29, 43, 72, 91, 405)

In addition, some figures were difficult to understand. Specifically, the ways to indicate significance were often extremely difficult to follow and I would consider other ways that they could be depicted. Some specific comments about the figures follow:

A) Fig. 8 colors should be switched. Red is typically associated with negative while green is typically associated with positive. Either that, or use an arrowhead like -> for positive and like -| for negative, as are often used with gene interaction maps.

Answer: Thanks for your comments. Fig. 8 colors have been switched. Red is associated with negative while green is associated with positive.

B) The change in labels from size fractions to a, b, c, d in Figs. 3-6 is quite confusing. I would suggest continuing to label with the size fraction numbers.

Answer: Thanks for your comments. Figs. 3-6 labels have been replaced to the size fraction numbers (>2 mm, 1-2 mm, 0.25-1 mm, <0.25 mm).

C) Significance indicators on Figs. 1, 2 and 6 were difficult to read and understand.

First, I would use commas to separate the significance between the different aggregates from the significance between the treatments, e.g. "ab,A" just to separate the two more. Second, the Shannon/Ace index significance indicators are exceedingly difficult to read. Is there another way that you can represent the significance in Fig. 6?

Answer: Thanks for your comments. Figs. 1-2 significance indicators have been revised adding commas. Fig. 6 significance indicators have been replaced to letters, e.g. "ab,A".

D) If you begin by clustering based on size fraction, I would continue to cluster by size fraction rather than switching to clustering by treatment halfway through.

Answer: Thanks for your comments. Both agronomic measures and aggregate size fraction affected soil microorganisms and carbon content. Fig. 1, Fig. 2 and Fig. 6 have been changed to cluster based on agronomic measures. In the RDA analysis, we mainly discussed the relationship between soil physicochemical properties and bacterial community composition in different particle sizes, which is a blank in previous research.

E) Fig. 5: What does M x A represent? The measures & aggregate size together?

Answer: Thanks for your comments. "M" means agronomic measures, "A" means aggregate size. We also have put it in the notes of Fig. 5.

I also have some methodological questions:

1) Can you comment on whether the qPCR primers are able to detect form II cbbL, or just form I? Given the high percentage of proteobacteria in these studies, it could be relevant to interpreting the results. If these primers are not able to detect form II cbbL, that should be noted in the discussion of the qPCR results as it will mean the Rubisco activity measurements and Rubisco abundance from qPCR are not measuring the same pool of Rubisco.

Answer: Thanks for your comments. The qPCR primers are able to detect form I cbbL. We have revised the method (Lines 177-178). We also have revised the discussion of RuBisCO activity and cbbL abundance in the manuscript (Lines 388-390). Form I occurs in all chemolithotrophs in the bacterial domain. Sequence analyses indicate that obligate lithotrophs

predominately possess form IA RuBisCO, while facultative lithotrophs predominately possess form IC. Previous studies have shown that the primers amplify both form IA and IC genes (Morsdorf et al., 1992; Tolli et al., 2005).

[1] Morsdorf, G., K. Frunzke, D. Gadkari, and O. Meyer. 1992. Microbial growth on carbon monoxide. *Biodegradation* 3:61-82.

[2] Tolli J, King GM. 2005. Diversity and Structure of Bacterial Chemolithotrophic Communities in Pine Forest and Agroecosystem Soils. *Appl Environ Microbiol* 71: 8411-8418.

2) The Rubisco activity units need more explanation. Is it $\text{nmol CO}_2 \cdot \text{h}^{-1} \cdot \text{g Rbc}^{-1}$? Or is the grams section referring to the soil solution? Activity is typically reported as $\mu \text{mol CO}_2 \cdot \text{min}^{-1} \cdot \text{mg Rbc}^{-1}$ or $\mu \text{mol CO}_2 \cdot \text{min}^{-1} \cdot \text{mol Rbc active site}^{-1}$.

Answer: Thanks for your comments. The grams section refers to the soil dry weight. We have revised RuBisCO activity units to $\text{nmol CO}_2 \text{ g}^{-1} \text{ soil h}^{-1}$ (Xiao et al., 2014). (Fig. 2)

[1] Xiao K, Bao P, Bao Q, Jia Y, Huang F, Su J, Zhu Y. 2014. Quantitative analyses of ribulose-1,5-bisphosphate carboxylase/oxygenase (RubisCO) large-subunit genes (cbbL) in typical paddy soils. *FEMS Microbiol Ecol* 87, 89-101

3) It is unclear if you quantified Rubisco content of the soil extracts, and you did not quantify how many active sites were functional, so you did not measure if the enzyme was getting more active. Likely, the "Rubisco activity" results represent an increase in the amount of active Rubisco present in the community, which is a different thing than increasing activity of the protein itself. I would like to see you mention this in your discussion to clarify the interpretation of the results, as the latter is far more likely than the former.

Answer: Thanks for your comments. At first, we extracted total soil protein. We did not measure if the RuBisCO was getting more active per protein. Maybe more active and increasing amount of Rubisco were occurred in some soil samples, but we don't need to explore it, because our main focus is on the overall RuBisCO activity per unit weight of soil (contribution to soil carbon fixation), whether it is due to increased enzyme activity (per protein) or more

enzyme quantity (Xiao et al., 2014; Wu et al., 2014).

[1] Xiao K, Bao P, Bao Q, Jia Y, Huang F, Su J, Zhu Y. 2014. Quantitative analyses of ribulose-1,5-bisphosphate carboxylase/oxygenase (RubisCO) large-subunit genes (cbbL) in typical paddy soils. *FEMS Microbiol Ecol* 87, 89-101

[2] Wu X, Ge T, Yuan H, Zhou P, Chen X, Chen S, Brookes P, Wu J. 2014. Evaluation of an optimal extraction method for measuring d-ribulose-1,5-bisphosphate carboxylase/oxygenase (RubisCO) in agricultural soils and its association with soil microbial CO₂ assimilation. *57*, 277-284.

4) I am not familiar with your structural equation modeling, and there were very few details in the Methods on how it was done, other than which R package was used - what settings did you use?

Answer: Thanks for your comments. We used RStudio (v4.3.1) packages “piecewiseSEM” and “ggplot2” (Line 202), and then, my data is read. My code was:

```
library(ggplot2)
library(piecewiseSEM)
getwd()
setwd("E:/data")
dt<-read.csv("SOC.SEM.csv",header =T)
dt
fit <- psem(
  lm(SOC~RubisCO+MBC+Shannon,dt),
  lm(MBC~cbbL+RubisCO+Shannon+Measures,dt),
  lm(cbbL~Shannon+Aggregates,dt),
  lm(RubisCO~Measures+Shannon+cbbL,dt),
  lm(Shannon~Measures+Aggregates,dt),
  lm(Aggregates~Measures,dt)
)
print(fit)
plot(fit)
summary(fit)
rsquared(lm(SOC~cbbL+RubisCO+MBC+Shannon+Ace+Measures+Aggregates,dt))
```

Line-by-line Comments:

1) Line 12: Abbreviating carbon to C is unnecessarily confusing when you are already going to abbreviate soil organic carbon and microbial biomass carbon to SOC and MBC. I would recommend using "carbon" when you are not using one of the other defined acronyms.

Answer: Thanks for your comments. We have changed the abbreviation for C to "carbon" in the revised manuscript.

2) Line 69-70: What is the relationship between SOC transformation, microbial communities, and enzyme activity? I'd prefer to see the results of the study written here instead of an indication that it has been studied.

Answer: Thanks for your comments. This sentence has been revised to "Previous reports suggest that agronomic measures affect SOC transformation, microbial communities, and enzyme activities, and these factors present dynamic interactions." (Lines 66-68)

3) Line 89: Would, or does? If this is not a hypothesis, "Microbial growth and function varies..." is more appropriate.

Answer: Thanks for your comments. This sentence has been revised to "Microbial growth and function vary in..." (Line 88)

4) Line 149: This method is typically called the "NADH-coupled spectrophotometric" method, as it does not involve fluorescence. Also, please cite the Wu et al. 2014 study in addition to the currently cited study, as that is the one where the method for assessing the Rubisco activity is described in detail.

Answer: Thanks for your comments. This method name has been described "NADH-coupled spectrophotometric method", and we have cited the Wu et al. 2014 study in revised manuscript. (Line 148)

5) Lines 366-369: What exactly are you referring to when you say nutrient and energy supply? Is it an assumption based on the previous study cited in this paragraph, or is it based on specific measured values in this study? If the former, it would be helpful if you reworded it to indicate that your results are consistent with the previous study instead of combining them into one sentence with no way to distinguish where the information

is coming from.

Answer: Thanks for your comments. We have revised it to “substrate supply”. Based on the previous study, NPK treatment decreased soil N, P, and K availability, as well as SOC content. This sentence has been revised to “there is low substrate supply in the poor soil structure following long-term NPK application (Li et al., 2022), which is consistent with our study.” (Lines 371-374)

[1] Li X, Li B, Chen L, Liang J, Huang R, Tang X, Zhang X, Wang C. 2022. Partial substitution of chemical fertilizer with organic fertilizer over seven years increases yields and restores soil bacterial community diversity in wheat–rice rotation. *Eur J Agron* 133: 126445.

6) Lines 390-392: I do not understand what this sentence is indicating, as it seems to contradict the sentence before it. Is the better predictor for overall carbon fixation dynamics cbbL or Rubisco activity? If the latter, I would suggest something like, "Thus Rubisco activity may be a better predictor for overall carbon fixation dynamics (fixation, release, storage) than the abundance of cbbL in the community."

Answer: Thanks for your comments. This sentence has been revised to “Thus, Rubisco activity may be a better predictor for overall carbon fixation dynamics (fixation, release, storage) than the abundance of cbbL in the community.” (Lines 396-398)

As reported by Trivedi et al. (2016), the results indicate that the soil microbial community regulates enzyme activity through the functional gene abundance. We also found that the functional gene cbbL is more sensitive to bacterial community, which in turn has consequences for RuBisCO activity directly linked to SOC.

[1] Trivedi P, Delgado-Baquerizo M, Trivedi C, Hu H, Anderson LC, Jeffries TC, Zhou J, Singh BK. 2016. Microbial regulation of the soil carbon cycle: evidence from gene–enzyme relationships. *ISME J* 10: 2593-2604.

7) I don't think section 3.2 currently depicts the Rubisco activity results in enough detail. Lines 224-225 state that Rubisco activity was increased significantly in the NPKS in the 0.25-1-mm aggregate class, but it also increased significantly in the <0.25 and 1-

2. I would suggest: "...was increased significantly in most size fractions, with the exception of the >2mm aggregates (Fig. 2a)." I also would like to see you discuss that the NPK treatment decreased Rubisco activity in a couple of the size fractions.

Answer: Thanks for your comments. This sentence has been revised to "Relative to the control, NPKS treatment significantly increased RuBisCO activity of aggregates across most size fractions, exception for >2 mm aggregate." (Lines 224-226)

We have discussed "the NPK treatment decreased Rubisco activity" in the revised manuscript (Lines 320-322). Long-term NPK and Ctrl treatments decreased MBC contents and RuBisCO activity in aggregates, resulting in a relatively low carbon turnover rate, which may reduce soil carbon assimilation capacity and destroy soil aggregate structure.

8) Lines 406-407: I don't think you have the support for this sentence as it is written. As stated above, regulation is not an appropriate word to use in this context. I would suggest something more like "Based on our model, remodeling of bacterial community structure in the .25-1-mm aggregate fraction increases *cbbL* abundance, which leads to increased Rubisco activity in the community. This in turn increases SOC..."

Answer: Thanks for your comments. This sentence has been revised to "Based on our model, remodeling of bacterial community structure in the 0.25–1 mm aggregate fraction increases *cbbL* abundance, which leads to increased Rubisco activity in the community. This in turn increases SOC and MBC content." (Lines 404-406)

Reviewer #3 (Comments for the Author):

This study exposes that straw application promotes soil carbon storage by regulating-associated bacterial community structure and RuBisCO activity. The purpose of the study is clear. However, the manuscript still has more problems in writing and basic formatting as follows:

1. Line 23: "Bacterial community richness and diversity....." is redundant because bacterial community diversity includes richness. Check and modify the whole

manuscript.

Answer: Thanks for your comments. This sentence has been revised to “Bacterial community diversity...” (Line 20). In results analysis, Shannon index represents diversity, Ace index represents richness. To avoid redundant, we have deleted “richness and” in discussion.

2. Line 96-100: The authors mentioned in hypothesis (2) that: the 0.25-1-mm aggregate fraction is the main site of C fixation, with greater microbial biomass, cbbL abundance, and RuBisCO activity, promoting C renewal. However, the introductory section only mentions the advantages of macroaggregates (>0.25 mm size) and does not point out the specificity of the size 0.25-1 mm, so such an assumption is unfounded.

Answer: Thanks for your comments. This hypothesis has been revised to “the >0.25 mm aggregate fractions are the main sites of carbon fixation” (Lines 97-98). In our study, >0.25 mm aggregate fractions play a crucial role in carbon fixation, especially 0.25-1 mm aggregate.

3. Use spaces rather than hyphens between values and units, e.g. 2-mm to 2 mm. Check and modify the whole manuscript.

Answer: Thanks for your comments. All aggregate size fractions have been used spaces in the revised manuscript.

4. Line 389: "As reported by Trivedi et al. (45), the results of the present study indicate that....", this sentence miscellaneous.

Answer: Thanks for your comments. This hypothesis has been revised to “As reported by Trivedi et al. (46), the results indicate that the soil microbial community regulates enzyme activity through functional gene abundance.” (Lines 395-396)

5. The manuscript has too many sentences like "in the present study" that are redundant, suggest deleting them and just writing the results. Check and revise the entire manuscript.

Answer: Thanks for your comments. "in the present study" and "of the present study" have been deleted in the revised manuscript.

6. Line 374-292: Is it better to change the bacterial community structure to bacterial community composition in part 3.5?

Answer: Thanks for your comments. We have revised “bacterial community structure” to “bacterial community composition” in part 3.5. (Lines 274-289)

7. Line 393-400: There is no discussion in this passage, but rather a presentation of the results of an SEM model, which is not quite appropriate for the author to place in the discussion section.

Answer: Thanks for your comments. The results of an SEM model have been transferred to part 3.5, explaining the relationships between soil carbon and bacterial community. (Lines 290-297)

8. The size of the font needs to be uniform across all figures. The use of black should be avoided in the colours of Figures_1 and Figures_2. Figure_4 lacks vertical labels. The statistical analyses in Figure_6 are better changed to multiple comparisons with one-way ANOVA, i.e., the differences between groups are replaced by a, b, c, etc., for visual clarity.

Answer: Thanks for your comments. The colours of Figures_1 and Figures_2 have been changed. Figure_4 was made in the Majorbio cloud platform (<https://cloud.majorbio.com>), without vertical labels. Figure_6 has been revised bar graph with one-way ANOVA.

9. Manuscripts need to be in Times New Roman font. Attention should be paid to basic formatting.

Answer: Thanks for your comments. “Arial” font is also used in most manuscripts. Arial font in our draft presents clearer.

Re: Spectrum00088-25R1 (Straw application promotes soil carbon storage by affecting aggregate-associated bacterial community structure and RuBisCO activity: A 35-year field experiment)

Dear Prof. Bing Li:

Thank you for the privilege of reviewing your work. Below you will find my comments, instructions from the Spectrum editorial office, and the reviewer comments.

Revision Guidelines

Sincerely,
Weimin Sun
Editor
Microbiology Spectrum

Reviewer #1 (Comments for the Author):

It is clear that the authors have made extensive changes to the manuscript in response to the comments, and their hard work is reflected in the improved quality of the manuscript. In particular, the changes to figures 1, 2, and 6 have much improved their readability, as have the changes to sections 3.1-3.4.

However, there are two pieces of feedback from the previous review that have not been addressed sufficiently yet.

1) Rubisco form II in chemolithoautotrophs: In lines 84-85, you state that "analyses of form I Rubisco offer the greatest insights for understanding chemolithotroph activity." You also stated in your response to reviewer comments that "Form I occurs in all chemolithotrophs in the bacterial domain. Sequence analyses indicate that obligate lithotrophs predominately possess form IA RuBisCO, while facultative lithotrophs predominately possess form IC. Previous studies have shown that the primers amplify both form IA and IC genes (Morsdorf et al., 1992; Tolli et al., 2005)." However, work from the last two decades has shown that many bacterial chemolithoautotrophs contain form II, some as their only form of Rubisco; see <https://doi.org/10.1093/jxb/erm297> for more information on some gene arrangements and overlap in Rubisco forms in bacteria. In addition, the fastest known Rubiscos are now form II, rather than form I. The fastest is a form II from a soil Gallionella; see <https://doi.org/10.15252/embj.2019104081> for more information. Please revise lines 84-85, lines 177-178, and lines 385-390 given these more recent studies.

2) The use of regulate as a verb: I appreciate the authors' diligence to revise the paper to remove the word regulating in most circumstances. However, there are still two places where "regulating" is used in the paper. As far as I can tell, there are two different explanations for the increased cbbL gene copy number: increased transcription of cbbL or to a higher percentage of the community being autotrophic. The data in the paper is not sufficient to determine if either of these options fully explains the increase, although the extensive community shifts presented here make it more likely to be the latter. Thus, "regulating" in lines 27 and 296 is not appropriate.

Some further comments:

- It would be helpful to define the abbreviations Ctrl, NPK, and NPKS in the first section of the results for future readers who may have skipped the methods.
- What does "these factors present dynamic interactions" in lines 67-68 mean?
- What does "influencing microorganisms promote SOC composition and decomposition" in lines 71-72 mean?
- Line 279: explained approximately 85.75% what? There's no noun associated with the percentages.
- I am confused what respectively refers to in lines 292-294. Is 83% SOC and 79% MBC? Or is 83% for both SOC & MBC related to agronomic measures & 79% for both SOC & MBC related to aggregate sizes?
- 369-370: Where was turnover rate measured?
- In lines 385-387: "cbbL was considered to indicate soil carbon-fixing potential". Later in lines 391-393 you "propose that RuBisCO activity serves as a more reliable indicator for autotrophic carbon sequestration potential in soil aggregates, relative to cbbL abundance." These two statements seem to contradict each other.

Reviewer #3 (Comments for the Author):

Fig.4 indicates what the values on the y-axis represent, whether it is a percentage of relative abundance or something else, adding labels, e.g., Relative abundance (%).

Also, your Fig.3 and 4 show the same data, correct? Fig.4 is just a difference analysis of the major phyla in Fig.3. In this case, firstly, the y-axis coordinates of Fig.3 and 4 are different and it is recommended that they be aligned, and secondly, the top six phyla of relative abundance in Fig.3 do not agree with Fig.4, so check and amend.

Response to Reviews 3.0

We would like to thank the editors and the reviewers for their insightful and constructive comments on our manuscript. We have carefully considered all the comments, and revised the manuscript accordingly. Below we provide a point-by-point response to each of the comments raised by the reviewers. The reviewers' comments are written in bold and our responses are in normal characters. The page and line numbers in our response correspond to those in the revised manuscript.

Reviewer #1 (Comments for the Author):

It is clear that the authors have made extensive changes to the manuscript in response to the comments, and their hard work is reflected in the improved quality of the manuscript. In particular, the changes to figures 1, 2, and 6 have much improved their readability, as have the changes to sections 3.1-3.4.

However, there are two pieces of feedback from the previous review that have not been addressed sufficiently yet.

1) Rubisco form II in chemolithoautotrophs: In lines 84-85, you state that "analyses of form I Rubisco offer the greatest insights for understanding chemolithotroph activity." You also stated in your response to reviewer comments that "Form I occurs in all chemolithotrophs in the bacterial domain. Sequence analyses indicate that obligate lithotrophs predominately possess form IA RuBisCO, while facultative lithotrophs predominately possess form IC. Previous studies have shown that the primers amplify both form IA and IC genes (Morsdorf et al., 1992; Tolli et al., 2005)." However, work from the last two decades has shown that many bacterial chemolithoautotrophs contain form II, some as their only form of Rubisco; see <https://doi.org/10.1093/jxb/erm297> for more information on some gene arrangements and overlap in Rubisco forms in bacteria. In addition, the fastest known Rubiscos are now form II, rather than form I. The fastest is a form II from a soil Gallionella; see <https://doi.org/10.15252/emj.2019104081> for more information. Please revise lines 84-85, lines 177-178, and lines 385-390 given these more

recent studies.

Answer: Thanks for your comments. Many bacterial chemolithoautotrophs contain form II, some as their only form of Rubisco, while measuring form - I rubiscos is more represented in soil samples compared to form - II. Badger et al., (2008) mentioned, “Terrestrial sites such as soils (Selesi et al., 2005; Tolli and King, 2005) and volcanic deposits (Nanba et al., 2004) have also been sampled. These reveal the presence of facultative and obligate chemolithotrophs. Agricultural and pine forest soils showed a narrow diversity of Form IA Rubiscos related to Nitrobacter species, and a high diversity of Form IC Rubiscos related to a range of alpha- and beta-proteobacteria (Selesi et al., 2005; Tolli and King, 2005). Volcanic deposits showed Form IC Rubisco related to a number of facultative chemolithotrophs from the alpha-proteobacteria. However, microbial mats overlying the volcanic material were dominated by Form IA Rubiscos related to Thiobacilli species (Nanba et al., 2004).” <https://doi.org/10.1093/jxb/erm297>

In the study of Davidi et al., (2020) mentioned, “Form - II rubisco from soil bacterium *Gallionella* sp. was found to have six - fold faster carboxylation rate than the median plant enzyme, and nearly two - fold faster than the fastest measured rubisco to date.” Davidi et al., (2020) also mentioned, “In accordance with their ubiquity in sequence databases, form - I rubiscos are also highly overrepresented in kinetic studies. Ninety - five percent of reported rubisco kinetic values are of form - I (80% are from plants), with only a handful of studies accounting for other forms (Flamholz et al, 2019; Jeske et al, 2019).” Thus, Davidi’s work focused on form-II and form-II/III rubiscos. Their work further expands the contribution of form - II, form - III, and form - II/III rubiscos to the global sequence diversity of this enzyme. <https://doi.org/10.15252/emj.2019104081> However, we think form - I rubiscos (97% of the sequenced rubiscos are form - I) is more represented in soil samples, compared to single but more active species (*Gallionella* sp.). In our work, limited progress has been made in promoting soil carbon storage, future metagenomic efforts will explore more rubiscos form.

We have revised them to “analyses of form I RuBisCO offer the greatest insights for understanding soil autotrophic bacteria activity” (Line 82-84); “The large subunit of form I RubisCO is encoded by the *cbbL* gene (17).” (Line 176).

2) The use of regulate as a verb: I appreciate the authors' diligence to revise the paper to remove the word regulating in most circumstances. However, there are still two places where "regulating" is used in the paper. As far as I can tell, there are two different explanations for the increased cbbL gene copy number: increased transcription of cbbL or to a higher percentage of the community being autotrophic. The data in the paper is not sufficient to determine if either of these options fully explains the increase, although the extensive community shifts presented here make it more likely to be the latter. Thus, "regulating" in lines 27 and 296 is not appropriate.

Answer: Thanks for your comments. We have revised "regulating" to "affecting" and "increasing", respectively. (Line 26 and 296)

Some further comments:

1) It would be helpful to define the abbreviations Ctrl, NPK, and NPKS in the first section of the results for future readers who may have skipped the methods.

Answer: Thanks for your comments. We have defined Ctrl, NPK, and NPKS in the first section of the results. (Line 204-205)

2) What does "these factors present dynamic interactions" in lines 67-68 mean?

Answer: Thanks for your comments. Soil organic carbon transformation, microbial communities, and enzyme activities are not acting in isolation, they influence and respond to each other in ways that change over time or under different conditions. We have revised this sentence to "....., which interact and interconnect". (Line 65-67)

3) What does "influencing microorganisms promote SOC composition and decomposition" in lines 71-72 mean?

Answer: Thanks for your comments. This sentence has been revised to "Substrates in soil aggregates are reportedly a primary factor influencing microbial activity that drives SOC formation and decomposition". (Line 70-71)

4) Line 279: explained approximately 85.75% what? There's no noun associated with the percentages.

Answer: Thanks for your comments. Together, RDA1 and RDA2 explained 85.75% of the total variation. We have revised this sentence to “the cumulative variance explained by RDA1 and RDA2 was 85.75%, 84.08%, 86.44%, and 65.42% for the four soil aggregates fractions (<0.25 mm, 0.25 – 1 mm, 1 – 2 mm, >2 mm), respectively (Fig. 7).” **(Line 278-280)**

5) I am confused what respectively refers to in lines 292-294. Is 83% SOC and 79% MBC? Or is 83% for both SOC & MBC related to agronomic measures & 79% for both SOC & MBC related to aggregate sizes?

Answer: Thanks for your comments. This is 83% of variation in SOC and 79% of variation in MBC. This sentence has been revised to “According to the results, agronomic measures and aggregate sizes directly or indirectly explained 83% of SOC variation and 79% of MBC variation, respectively, by influencing soil biochemical properties and bacterial attributes.” **(Line 292-294)**

6) 369-370: Where was turnover rate measured?

Answer: Thanks for your comments. We haven't measured turnover rate. Enzyme activity and MBC can reflect turnover rate to some extent. Combined with our data, this sentence has been revised “Therefore, the contents of SOC and MBC driven by microbial activity in the 0.25 – 1 mm aggregate fraction were higher than those in the <0.25 mm and >2 mm fractions.” **(Line 370-371)**

7) In lines 385-387: "cbbL was considered to indicate soil carbon-fixing potential". Later in lines 391-393 you "propose that RuBisCO activity serves as a more reliable indicator for autotrophic carbon sequestration potential in soil aggregates, relative to cbbL abundance." These two statements seem to contradict each other.

Answer: Thanks for your comments. Previous report suggested that “*cbbL* was considered to indicate soil carbon-fixing potential (Lynn et al., 2017)”. Based on this, in soil aggregates, we found RuBisCO activity is a more reliable indicator for autotrophic carbon sequestration potential, relative to *cbbL* abundance. We have deleted this ambiguous sentence “Based on

the strong positive correlation between soil cbbL abundance and RuBisCO activity, cbbL was considered to indicate soil carbon-fixing potential (45).” (Line 387)

Lynn T, Ge T, Yuan H, Wei X, Wu X, Xiao K, Kumaresan D, Yu SS, Wu J, Whiteley AS. 2017. Soil Carbon-Fixation Rates and Associated Bacterial Diversity and Abundance in Three Natural Ecosystems. *Microb Ecol* 73: 645-657.

Reviewer #3 (Comments for the Author):

Fig.4 indicates what the values on the y-axis represent, whether it is a percentage of relative abundance or something else, adding labels, e.g., Relative abundance (%).

Answer: Thanks for your comments. We have revised the y-axis of Fig. 4 to “Relative abundance”, which is same with Fig. 3.

Also, your Fig.3 and 4 show the same data, correct? Fig.4 is just a difference analysis of the major phyla in Fig.3. In this case, firstly, the y-axis coordinates of Fig.3 and 4 are different and it is recommended that they be aligned, and secondly, the top six phyla of relative abundance in Fig.3 do not agree with Fig.4, so check and amend.

Answer: Thanks for your comments. ①Fig.3 and 4 showed the same data, we have revised the y-axis of Fig. 4 to “Relative abundance”, which is same with Fig. 3. ②Figure 4 is the comparison of the top 6 relative abundance of bacterial phyla with significant differences. We have revised manuscript to “Multigroup comparison was performed to compare the relative abundances of bacterial phyla showing significant differences under different agronomic measures and aggregate fractions (Fig. 4).” (Line 248-250)

“Fig. 4 Relative abundance of bacterial phyla showing significant differences in soil aggregates under different management measures (Only the top six with significant differences are shown).”

(The note of Fig. 4, line 572-573)

Re: Spectrum00088-25R2 (Straw application promotes soil carbon storage by affecting aggregate-associated bacterial community structure and RuBisCO activity: A 35-year field experiment)

Dear Prof. Bing Li:

Thank you for the privilege of reviewing your work. Below you will find my comments, instructions from the Spectrum editorial office, and the reviewer comments.

I have carefully reviewed the manuscript without sending out for another round of review. Here is a summary of the changes for further improvements.

Summary of revisions incorporated

The authors have revised the text to clarify that form I Rubisco is more representative in soil samples, while acknowledging the presence and activity of form II in certain bacteria (e.g., *Gallionella* sp.). Updated lines 82-84 and 176 to reflect this.

Replaced "regulating" with "affecting" (Line 26) and "increasing" (Line 296).

Revised "these factors present dynamic interactions" to "which interact and interconnect" (Lines 65-67).

Clarified "influencing microorganisms promote SOC composition and decomposition" to a more precise statement about microbial activity driving SOC formation/decomposition (Lines 70-71).

Added context for the 85.75% variance explained by RDA1 and RDA2 (Lines 278-280).

Specified that 83% refers to SOC variation and 79% to MBC variation (Lines 292-294).

Clarified that turnover rate was not directly measured but inferred from enzyme activity and MBC (Lines 370-371).

Removed the ambiguous sentence about *cbbL* indicating carbon-fixing potential (Line 387) and emphasized Rubisco activity as a more reliable indicator.

Aligned y-axis labels ("Relative abundance") and clarified that Fig. 4 shows only the top six phyla with significant differences (Lines 248-250, 572-573).

Suggestions for further refinements

While the revisions address the reviewer's concerns, the manuscript could further emphasize the ecological significance of form II Rubisco (e.g., its faster carboxylation rate) and its potential implications for carbon fixation in specific environments.

The SEM (Fig. 8) could be better explained in the text, particularly the negative path coefficient between bacterial community and MBC content, which seems counterintuitive. A brief justification would be helpful.

The term "autotrophic carbon sequestration potential" (Line 391) should be clarified to distinguish between potential (gene abundance) and realized activity (enzyme activity).

Conclusion

The authors have carefully addressed the reviewers' comments. The changes have been included in the combined document.

Small clarifications and checks for consistency, such as figure labels and SEM explanations, would further improve the paper.

No major issues were observed. The paper is now clearer and stronger in its argument.

Revision Guidelines

Data availability: ASM policy requires that data be available to the public upon online posting of the article, so please verify all links to sequence records, if present, and make sure that each number retrieves the full record of the data. If a new accession

number is not linked or a link is broken, provide Spectrum production staff with the correct URL for the record. If the accession numbers for new data are not publicly accessible before the expected online posting of the article, publication may be delayed; please contact production staff (Spectrum@asmusa.org) immediately with the expected release date.

Sincerely,
Weimin Sun
Editor
Microbiology Spectrum

Response to Reviews 4.0

We would like to thank the editors' insightful and constructive comments on our manuscript. We have carefully considered all the comments, and revised the manuscript accordingly. Below we provide a point-by-point response to each of the comments raised by the editors. The editors' comments are written in bold and our responses are in normal characters. The page and line numbers in our response correspond to those in the revised manuscript.

Suggestions for further refinements

1) While the revisions address the reviewer's concerns, the manuscript could further emphasize the ecological significance of form II Rubisco (e.g., its faster carboxylation rate) and its potential implications for carbon fixation in specific environments.

Answer: Thanks for your comments. We have emphasized the ecological significance of form II Rubisco in **L 399-403**

"In our work, limited progress has been made in promoting soil carbon storage. Future metagenomic efforts will explore more RuBisCO form, particularly Form II RuBisCO, which has been found to operate twice as fast as the fastest carboxylation rate measured RuBisCO to date (46). The potential implications of Form II RuBisCO for carbon fixation have likely been underestimated."

2) The SEM (Fig. 8) could be better explained in the text, particularly the negative path coefficient between bacterial community and MBC content, which seems counterintuitive. A brief justification would be helpful.

Answer: Thanks for your comments. We have clarified this point in **L 362-366**

"Unexpectedly, MBC content showed a negative correlation with bacterial community diversity (Fig. 8). This pattern likely reflects the dominant role of microbial life-history strategy selection (k/r-strategists) in aggregate, suggesting decreased microbial carbon use efficiency or the dominance of bacterial taxa contributing less to MBC."

3) The term "autotrophic carbon sequestration potential" (Line 391) should be clarified to distinguish between potential (gene abundance) and realized activity (enzyme activity).

Answer: Thanks for your comments. We have revised it to "we propose that RuBisCO activity better reflects the realized autotrophic carbon sequestration capacity in soil aggregates, relative to *cbbL* abundance." **L393-394**

Re: Spectrum00088-25R3 (Straw application promotes soil carbon storage by affecting aggregate-associated bacterial community structure and RuBisCO activity: A 35-year field experiment)

Dear Prof. Bing Li:

The authors have satisfactorily addressed all comments, and the manuscript is now ready for further editorial processing by the journal.

Your manuscript has been accepted, and I am forwarding it to the ASM production staff for publication. Your paper will first be checked to make sure all elements meet the technical requirements. ASM staff will contact you if anything needs to be revised before copyediting and production can begin. Otherwise, you will be notified when your proofs are ready to be viewed.

Sincerely,
Weimin Sun
Editor
Microbiology Spectrum